# Robust Explanations of Graph Neural Networks via Graph Curvatures

**Yazheng Liu[1], Xi Zhang[2], Sihong Xie*[1], Hui Xiong[1]**
[1] The Hong Kong University of Science and Technology (Guangzhou), Guangzhou, China
[2] The Beijing University of Posts and Telecommunications, Beijing, China
yliu533@connect.hkust-gz.edu.cn, zhangx@bupt.edu.cn
{sihongxie,xionghui}@hkust-gz.edu.cn,

## Abstract

Explaining graph neural networks (GNNs) is a key approach to improve the trustworthiness of GNN in high-stakes applications, such as finance and healthcare. However, existing methods are vulnerable to perturbations, raising concerns about explanation reliability. Prior methods enhance explanation robustness using model retraining or explanation ensemble, with certain weaknesses. Retraining leads to models that are different from the original target model and misleading explanations, while ensemble can produce contradictory results due to different inputs or models. To improve explanation robustness without the above weaknesses, we take an unexplored route and exploit the two edge geometry properties curvature and resistance to enhance explanation robustness. We are the first to prove that these geometric notions can be used to bound explanation robustness. We design a general optimization algorithm to incorporate these geometric properties into a wide spectrum of base GNN explanation methods to enhance the robustness of base explanations. We empirically show that our method outperforms six base explanation methods in robustness across nine datasets spanning node classification, link prediction, and graph classification tasks, improving fidelity in 80% of the cases and achieving up to a 10% relative improvement in robust performance. The code is available at https://github.com/yazhengliu/Robust_explanation_curvature.

## 1 Introduction

Graph neural networks (GNNs) have proven their strengths in many real world applications, such as social network modeling [1] and fraud detection [2]. To establish human trust in GNN prediction, it is essential to make GNN predictions transparent to humans in risk-critical applications [3, 4]. For example, biological researchers may seek to understand the reasoning behind GNN predictions for protein function to verify whether the model is trustworthy. Many explanation methods have been designed to explain the behavior of GNNs.

However, recent studies have shown that these explanation methods are vulnerable to perturbations [5, 6, 7]. The fragility of explanations [8] can mislead users into making wrong decisions, raising security concerns in high-stakes domains such as finance, healthcare, and criminal justice [9, 10]. For example, we consider a graph in which nodes represent customers and edges denote financial relationships such as borrowing or transaction flows. We use GNNs to predict personal loan approvals. If a loan is denied, customers may feel the need to understand the possible reasons and corrections to improve the chance of future approval. When small changes in the graph structure (adding or removing a transaction edge) lead to substantially different explanations, trust in the model may be undermined as customers begin to question the decisions of companies.

---

*Sihong Xie is the the corresponding author.

39th Conference on Neural Information Processing Systems (NeurIPS 2025).

Most methods for robust GNN explanations focus on the robust evaluation of explanations [11, 12] and robust counterfactual explanations [13, 14]. Robust evaluation methods introduce new metrics to compute the fidelity [15] of explanations while accounting for distribution shifts during the evaluation process. Robust counterfactual explanation methods aim to extract explanations similar to the input graph that flip the model prediction and keep stable under edge perturbations. Fewer works focus on the robustness of explanations [16]. One work define robustness of explanations as the expected change in predicted probabilities between the original and perturbed graphs, with the explanation subgraph fixed during perturbations [17]. We adopt this definition, formally stated in Equation (2). To improve the robustness of explanations, one approach is to retrain models with an extra regularization term [18]. They theoretically and empirically illustrate that the retrained model has improved explanation robustness. However, these retrained models are different from the original model, so that the robust explanations in fact **do not** explain the original model (Figure 1 (b) top). Alternatively, ensemble methods improve explanation robustness. For example, [19] propose to combine explanations from different algorithms, while others [16] generate perturbed graphs and aggregate explanations from these perturbed graphs. However, as target models can be sensitive to small perturbations, multiple explanations can disagree on graph element importance, leading to uncertainty and sabotaging the trustworthiness of the aggregated explanations (Figure 1 (b) bottom).

To address these challenges, we leverage two geometric properties of graph edges to generate robust explanations without model retraining or explanation ensembling. In particular, two important concepts in graph theory are *Ricci curvature* and *effective resistance*. The Ollivier-Ricci curvature [20] and effective resistance [21] can measure the connectivity between the neighborhoods of nodes $u$ and $v$. Positive curvature and low effective resistance indicate efficient message transmission between neighborhoods, while negative curvature and high effective resistance suggest that edge $(u, v)$ acts as a bottleneck of message transmission [22]. Recent works have shown that these properties are related to over-squashing in GNNs [23, 24, 25]. Nevertheless, there has been no work exploring the relationship between graph geometry and explanation robustness.

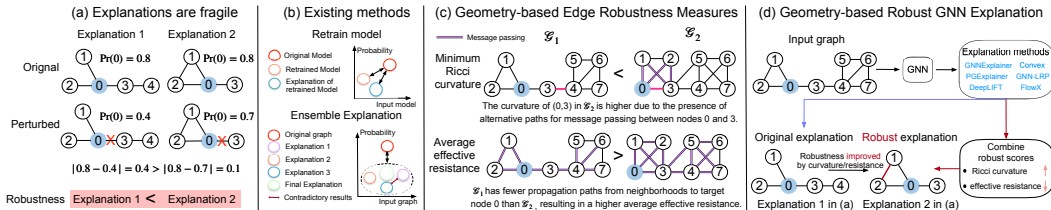

Figure 1: Assuming a GNN model has 3 layers, we use node classification as an example. (**a**): the explanations become fragile if the edge (0,3) is removed. (**b**): significant differences between the retrained and original models make the robust explanations not faithful to the original model. For ensemble explanations, contradictory results can lead to more uncertain explanations. (**c**): the prediction of node 0 on $\mathcal{G}_2$ is more robust than that on $\mathcal{G}_1$, as removing edge (0,3) in $\mathcal{G}_1$ significantly affects the model's predictions. We prove that higher minimum Ricci curvature and lower average effective resistance contribute to greater robustness of model predictions. While edge (3,4) in $\mathcal{G}_1$ and edge (0,3) in $\mathcal{G}_2$ have the smallest Ricci curvatures in their respective graphs, the curvature of (0,3) in $\mathcal{G}_2$ is higher due to the presence of alternative paths for message passing between nodes 0 and 3. The $\mathcal{G}_1$ has fewer propagation paths from neighborhoods to target node 0 than $\mathcal{G}_2$, resulting in a higher average effective resistance. (**d**): we use Ricci curvature and effective resistance as robust scores and combine them with importance scores from explanation methods to obtain robust explanations.

We first theoretically analyze that the Ollivier-Ricci curvature and effective resistance can be used to bound the robustness of model prediction and explanations, as shown in Theorems 4.3, 4.4, 4.7 and 4.8. Specifically, higher minimum Ollivier-Ricci curvature and lower average effective resistance across all edges in the graph and explanation subgraphs are associated with greater robustness in model predictions (Figure 1 (c)) and explanations. Based on these theoretical results, as shown in Figure 1 (d), we leverage Ricci curvature and effective resistance as robust scores to enhance explanation reliability and our method is agnostic to explanation methods. We propose objective functions that improve the robustness of existing explanation methods without modifying the model or generating multiple potentially contradicting explanations. Empirically, on nine datasets across node classification, link prediction, and graph classification tasks, we demonstrate that our method improves robustness in six state-of-the-art GNN explanation methods, enhancing fidelity in 80% of the cases and achieving up to a 10% relative improvement in robustness.

## 2   Related Work

**Explainability and explanation robustness.** Recent post-hoc explanation methods for GNNs can be categorized into instance-level and model-level methods [15]. Instance level explanation methods [3, 4, 26, 27, 28, 29] focus on explaining model predictions by identifying important subgraphs that strongly correlate with predictions. For example, GNNexplainer [3] and PGExplainer [27] learn edge masks to identify important edges by maximizing the mutual information to explain the prediction. Methods like GNN-LRP [4], DeepLIFT [26], FlowX [28] and Convex [29] analyze the importance of graph walks. Model level explanation methods [30] produce a high-level explanation about the general behaviors of GNNs. However, these methods are vulnerable to small adversarial perturbations [6, 7, 8]. Some studies aim to enhance explanation robustness through adversarial training [18, 31, 32]. The retrained model may be different from the original model, thus the explanations may not faithfully reflect the model [17]. Unlike training-based approaches, researchers have proposed explanation ensemble methods to enhance explanation robustness. For example, the explanation averaged across different methods improves robustness [19]. Besides, some researchers obtain robust explanations by averaging explanations of perturbed graphs [16]. However, results from different explanations may be contradictory, making it difficult to integrate them and potentially leading to less stable explanations.

**Graph curvature.** Efforts have been made to extend the geometric concept of curvature to graphs [33, 34, 21]. Among these, the Ollivier-Ricci curvature [20] and effective resistance are prominent approaches [35]. Studies have demonstrated their relevance to the issues of over-squashing and over-smoothing in GNNs [22, 21, 23, 25]. For example, a rewiring technique has been proposed to alleviate over-squashing by increasing the curvature of edges in the graph [23]. Effective resistance has been utilized to quantify over-squashing in GNNs [21]. Furthermore, a theoretical analysis has linked both over-smoothing and over-squashing to Ollivier-Ricci curvature [22]. Moreover, they have been applied in complex networks [36, 37, 38] and deep clustering [39]. However, recent research has not explored their potential for enhancing the robustness of explanations.

## 3   Preliminaries

### 3.1   Notation

Let $\mathcal{G} = (\mathcal{V}, \mathcal{E})$ denote the connected, undirected, unweighted graph with the vertex set $\mathcal{V}$ and the edge set $\mathcal{E}$. Let $A$ and $D$ be the adjacency and degree matrices, respectively, with the Laplacian defined as $L = D - A$. The normalized adjacency matrix is $\hat{A} = D^{-1/2}AD^{-1/2}$ and the normalized Laplacian is $\hat{L} = I - \hat{A} = D^{-1/2}LD^{-1/2}$. Let $\mathcal{G}' = (\mathcal{V}', \mathcal{E}')$ be the perturbed graph generated by adding and/or removing edges in $\mathcal{G}$, where $\mathcal{V}' \subseteq \mathcal{V}$. Let $\mathcal{G}_s = (\mathcal{V}_s, \mathcal{E}_s)$ denote the explanation graph, where $\mathcal{E}_s \subseteq \mathcal{E}, \mathcal{V}_s \subseteq \mathcal{V}$. We use $u, v, p,$ and $q$ to denote nodes in a graph. We denote $u \sim v$ if edge $(u, v) \in \mathcal{E}$. Let $\mathcal{N}_u$ denote the 1-hop neighborhood of node $u$. The shortest path distance between two nodes $u, v$ is denoted by $d(u, v)$. $n, m,$ and $n', m'$ denote the degrees of node $u, v$ in $\mathcal{G}$ and $\mathcal{G}'$.

### 3.2   Graph Neural Networks

Let $\boldsymbol{h}(\mathcal{G})$ denote the node feature matrix of graph $\mathcal{G}$ and $\boldsymbol{h}^t(\mathcal{G})$ be the hidden vector at layer $t$ ($t = 1, \cdots, T$), with the convention that $\boldsymbol{h}^0(\mathcal{G}) = \boldsymbol{h}$. The hidden vector of node $u$ at layer $t$ is denoted by $\boldsymbol{h}_u^t(\mathcal{G})$, and is exactly the $u$-th row of $\boldsymbol{h}^t(\mathcal{G})$. The formulation for GNN is given by [22]

$$\boldsymbol{h}_u^{t+1}(\mathcal{G}) = \phi_t \left( \bigoplus_{v \in \mathcal{N}_u} \psi_t(\boldsymbol{h}_v^t(\mathcal{G})) \right), \tag{1}$$

where $\psi_t$ is a message function (e.g., a linear function), and $\phi_t$ is an update function (e.g., an activation function). $\bigoplus$ is an aggregating function. If $\bigoplus$ is the sum function, $\bigoplus = \sum_{v \in \mathcal{N}_u} A_{uv}$. If $\bigoplus$ is the mean function, $\bigoplus = \sum_{v \in \mathcal{N}_u} \frac{1}{|\mathcal{N}_u|}$.

For node classification, $\boldsymbol{h}_u^T(\mathcal{G})$ is mapped to the class distribution $\mathsf{Pr}_u(\mathcal{G})$ using the softmax or sigmoid function. In link prediction, we concatenate $\boldsymbol{h}_u^T(\mathcal{G})$ and $\boldsymbol{h}_v^T(\mathcal{G})$ as the input to a linear layer to obtain the logits: $\boldsymbol{h}_{uv}(\mathcal{G}) = \left\langle \left[ \boldsymbol{h}_u^T(\mathcal{G}); \boldsymbol{h}_v^T(\mathcal{G}) \right], \boldsymbol{\theta}_{LP} \right\rangle$, where $\boldsymbol{\theta}_{LP}$ is the parameter in linear layer. Then, it is mapped to the probability $\mathsf{Pr}_{uv}(\mathcal{G})$ of the edge $(u, v)$ existing using the sigmoid function.

For the graph classification task, average pooling of $\boldsymbol{h}_u^T$ across all nodes in graph $\mathcal{G}$ yields a single vector representation, which is used for classification and the probability is denoted as $\Pr(\mathcal{G})$.

# 4 Relating robustness to curvature and resistance

We assume that the update function $\phi_t$ is $L$-Lipschitz, i.e. $|\phi_t(\boldsymbol{x}_1) - \phi_t(\boldsymbol{x}_2)| \leq L|\boldsymbol{x}_1 - \boldsymbol{x}_2|, \forall \boldsymbol{x}_1, \boldsymbol{x}_2$. The softmax function is $L_1$-Lipschitz. The message function $\psi_t(\boldsymbol{x})$ is bounded, i.e. $|\psi_t(\boldsymbol{x})| \leq M|\boldsymbol{x}|, \forall \boldsymbol{x}, t \geq 1$. Additionally, $\left|\boldsymbol{h}_p^t\right| \leq C$ for all $p \in \mathcal{V}$ and $t \geq 0$. Let $|\cdot|$ denote the $L_2$ norm and $N = |\mathcal{V}|$. These notations will be used in the following lemma, proposition, and theorem.

**Definition 4.1.** Given an input graph $\mathcal{G}$ and explanation $\mathcal{G}_s = (\mathcal{V}_s, \mathcal{E}_s)$, we construct a perturbed graph $\mathcal{G}' = (\mathcal{V}', \mathcal{E}')$ by adding and/or deleting edges not in $\mathcal{E}_s$. For node $u$, the approximate robustness of explanation on the node classification task is defined as [40]:

$$\delta^* = \mathbb{E}_{\mathcal{G}'}|\Pr_u(\mathcal{G})_c - \Pr_u(\mathcal{G}')_c| \text{ s.t. } c = \arg\max_i \Pr_u(\mathcal{G})_i, \mathcal{E}_s \subseteq \mathcal{E}' \tag{2}$$

where $c$ denotes the predicted class of original graph $\mathcal{G}$. $\Pr_u(\mathcal{G})_i$ and $\Pr_u(\mathcal{G}')_i$ denote the probability of $u$ on graph $\mathcal{G}$ and $\mathcal{G}'$ for a given class $i$. A lower value indicates a greater robustness of the explanation. The definitions for link prediction and graph classification are provided in Appendix A.

## 4.1 Robustness and Ollivier-Ricci Curvature

**Ollivier-Ricci Curvature on Graphs.** The Ollivier-Ricci curvature [20] considers random walks from nearby points. Let $\mu_u(k)$ represent the probability that a random walker starting at $u$ reaches node $k$ after a certain number of steps. On a graph, the random walk $\mu$ is defined by [22]

$$\mu_u(k) = \begin{cases} \frac{1}{\deg(u)} & \text{if } k \sim u, \\ 0 & \text{otherwise.} \end{cases} \tag{3}$$

Then, for any $u \sim v$, the 1-Wasserstein distance $W_1(\mu_u, \mu_v)$ is given by

$$W_1(\mu_u, \mu_v) = \inf_{\pi \in \Pi(\mu_u, \mu_v)} \left( \sum_{(p,q) \in \mathcal{V}^2} \pi(p,q) d(p,q) \right),$$

where $\Pi(\mu_u, \mu_v)$ is the family of joint probability distributions of $\mu_u$ and $\mu_v$. The 1-Wasserstein distance measures the minimal cost required to transform $\mu_u$ into $\mu_v$. The Ollivier-Ricci curvature $\kappa(u, v)$ is defined as

$$\kappa(u, v) = 1 - \frac{W_1(\mu_u, \mu_v)}{d(u, v)}. \tag{4}$$

Ricci curvature measures how easily information propagates from $u$ to $v$ based on topological connections of different lengths of paths around edge $(u, v)$ [41, 42]. Negative curvature implies the edge is a "bottleneck", while positive curvature indicates the edge is present in a highly connected community. We use Ricci curvature to bound the robustness of model predictions and explanations.

**Proposition 4.2.** *Given any node $u$ in $\mathcal{G}$, assume that there exists a constant $\alpha > 0$ such that for edge $(u, v) \in \mathcal{E}$, the curvature is bounded by $\kappa(u, v) \geq \alpha > 0$. Let $\beta = \max(n, n')$. If $\bigoplus$ is the mean operation, $g_1(\alpha) = \frac{3(1-\alpha)}{\alpha}$. If $\bigoplus$ is the sum operation, $g_1(\alpha) = 2(1 - \alpha)$. Then,*

$$\left|\boldsymbol{h}_u^t(\mathcal{G}) - \boldsymbol{h}_u^t(\mathcal{G}')\right| \leq \beta LCM\left(4 + g_1(\alpha)\right). \tag{5}$$

Proposition 4.2 shows that for node $u$, the difference in the hidden representation vectors between $\mathcal{G}$ and $\mathcal{G}'$ is related to $\alpha$, the minimum value of $\kappa(u, v)$ for any $v$ such that $u \sim v$. $\alpha$ is specific to node $u$. Intuitively, a large $\alpha$ implies high curvature for all edges, suggesting efficient information flow between $u$ and its neighbors. Even with random perturbations, multiple alternative paths remain, leading to minimal change in representation of $u$. However, when $\alpha$ is small, some neighbor $v$ may have low curvature $\kappa(u, v)$, implying sparse connectivity. If edge $(u, v)$ is removed, information flow may be severely disrupted, increasing the change in node representation $\boldsymbol{h}_u^t(\mathcal{G})$. The proof is in Appendix C.1. We can use this proposition to bound the robustness of model predictions.

**Theorem 4.3** (Ricci curvature bounds the robustness of model prediction). *In a GNN with $T$ layers, let $\mathcal{G}_u = (\mathcal{V}_u, \mathcal{E}_u)$ denote the $T$-hop subgraph of target node $u$, where $\mathcal{V}_u = \{v \in \mathcal{V} | d(u,v) \leq T\}$. Let $\beta'$ be the maximum degree of nodes in the $T$-hop subgraph of node $u$ for both the original graph $\mathcal{G}$ and the perturbed graph $\mathcal{G}'$. Assume that there exists a constant $\alpha' > 0$ such that for edges $(u,v) \in \mathcal{E}_u$, the curvature is bounded by $\kappa(u,v) \geq \alpha' > 0$. Let $\eta = L_1 L^T M^T C$. Then, the following inequality holds for the node classification task,*

*If $\bigoplus$ is the sum operation,*

$$|\mathsf{Pr}_u(\mathcal{G}) - \mathsf{Pr}_u(\mathcal{G}')| \leq \eta\bigg( \big(1 + \beta'\big)^{T-1}\big(2\beta'(1 - \alpha') + 4\beta'\big) + \sum_{i=2}^{T} 2(\beta')^i\big(1 + \beta'\big)^{T-i}\bigg).$$

*If $\bigoplus$ is the mean operation,*

$$|\mathsf{Pr}_u(\mathcal{G}) - \mathsf{Pr}_u(\mathcal{G}')| \leq \eta\bigg( 2^{T-1}\big(4\beta' + \frac{3(1 - \alpha')}{\alpha'}\big) + 4(T - 1)\beta'\bigg).$$

Theorem 4.3 states that for a target node $u$, the robustness of model prediction is bounded by minimum $\kappa(u,v), \forall(u,v) \in \mathcal{E}_u$. A high $\alpha'$ indicates efficient message passing across all edges in $\mathcal{G}_u$, leading to robust predictions under perturbations. In contrast, a low $\alpha'$ shows that the presence of an edge $(u,v)$ with small $\kappa(u,v)$, which may bottleneck message propagation. Deleting such an edge can cause a significant change in predictions. The proof is provided in Appendix C.2. The link prediction task and the graph classification task can be seen in Appendix B.1 and B.2.

**Theorem 4.4** (Ricci curvature bounds the robustness of model explanations). *For node $u$, let $\mathcal{G}_s = (\mathcal{V}_s, \mathcal{E}_s)$ be the explanation subgraph of $\mathsf{Pr}_u(\mathcal{G})$. Let $\mathcal{G}' = (\mathcal{V}', \mathcal{E}'), \mathcal{E}_s \subseteq \mathcal{E}'$ denote perturbed graph by adding and/or deleting edges not in $\mathcal{E}_s$. Let $\alpha_s$ denote the minimum $\kappa(u,v)$, such that for edges $(u,v) \in \mathcal{E}_s$, the curvature is bounded by $\kappa(u,v) \geq \alpha_s > 0$. $\eta = L_1 L^T M^T C$. Let $\beta'$ be the maximum degree of nodes in the $T$-hop subgraph of node $u$ for both the original graph $\mathcal{G}$ and the perturbed graph $\mathcal{G}'$, and $\beta_s$ be the maximum degree of nodes in explanation graph $\mathcal{G}_s$. Then, for node classification task,*

*If $\bigoplus$ is the sum operation,*

$$|\mathsf{Pr}_u(\mathcal{G}) - \mathsf{Pr}_u(\mathcal{G}')| \leq \eta\bigg( \big(1 + \beta_s\big)^{T-1}\big(2\beta_s(1 - \alpha_s) + 4\beta_s\big) + \sum_{i=2}^{T} (2\beta'^2 + 2)^i\big(1 + \beta_s\big)^{T-i}\bigg).$$

*If $\bigoplus$ is the mean operation,*

$$|\mathsf{Pr}_u(\mathcal{G}) - \mathsf{Pr}_u(\mathcal{G}')| \leq \eta\bigg( 2^{T-1}\big(4\beta_s + \frac{3(1 - \alpha_s)}{\alpha_s}\big) + 4(T - 1)\beta' + 2(T - 1)\bigg).$$

Theorem 4.4 shows that the robustness of explanations is bounded by the minimum Ollivier-Ricci value of edges in the explanation subgraphs. A larger $\alpha_s$ allows messages to be transmitted efficiently across all edges, resulting in robust explanations. In this case, graph perturbations lead to minimal changes in the model's explanation. The proof is in Appendix C.3. The link prediction and graph classification tasks are in Appendix B.3 and B.4. According to the Theorem 4.3 and 4.4, we establish a relationship between Ollivier-Ricci curvature and robustness.

## 4.2 Robustness and Effective Resistance

Effective resistance quantifies how well two nodes are connected [24]. While many studies have explored its relationship with over-squashing, we aim to investigate its connection to the robustness of model predictions and explanations. Let $u$ and $v$ be vertices of $\mathcal{G}$. The ***effective resistance*** between $u$ and $v$ is defined

$$R_{u,v} = (1_u - 1_v)^\top L^+ (1_u - 1_v), \tag{6}$$

where $1_u, 1_v$ are the indicator vector of the vertices $u, v$ and $L^+$ is the pseudoinverse of $L$. The effective resistance can also be computed using the normalized Laplacian $\hat{L}$. When multiple paths exist between two nodes, the effective resistance $R_{u,v}$ is small, indicating higher connectivity. Conversely, when available paths between two nodes are limited, $R_{u,v}$ becomes larger, reflecting lower connectivity.

**Lemma 4.5.** *Let $G$ be a connected graph. Let $u$ and $v$ be two vertices. Then*

$$R_{u,v} = \left( \frac{1}{\sqrt{d_u}} 1_u - \frac{1}{\sqrt{d_v}} 1_v \right)^\top \hat{L}^+ \left( \frac{1}{\sqrt{d_u}} 1_u - \frac{1}{\sqrt{d_v}} 1_v \right),$$

*where $\hat{L}^+$ is pseudoinverse of the normalized Laplacian $\hat{L}$.*

**Lemma 4.6.** *Let $G$ be a connected, non-bipartite graph. Then $\hat{L}^+ = \sum_{j=0}^{\infty} \hat{A}_r^j$ [21].*

**Theorem 4.7** (Effective resistance bounds the robustness of model prediction). *Given a GNN with $T$ layers, for $u$, let $\mathcal{G}_u = (\mathcal{V}_u, \mathcal{E}_u)$ denote the $T$-hop subgraph of $u$. Let $\bar{R}_u = \frac{\sum_{(q,v) \in \mathcal{E}_u} R_{q,v}}{|\mathcal{E}_u|}$ denote the average effective resistance of edges in $\mathcal{G}_u$. Let $\gamma$ denote the maximum eigenvalue of $\hat{A}$, and $d_{\min} = \min_{v \in \mathcal{V}_u} \deg(v)$. Let $\eta = L_1 L^T M^T C$. Whether $\bigoplus$ is the mean or sum operation, the following inequality holds for the node classification task,*

$$|\mathrm{Pr}_u(\mathcal{G}) - \mathrm{Pr}_u(\mathcal{G}')| \le \eta \left( 2N + \bar{R}_u + \frac{2}{d_{\min}(1 - \gamma)} \right).$$

Theorem 4.7 shows that the robustness of the model is related to $\bar{R}_u$. A smaller $\bar{R}_u$ indicates greater robustness. The $\bar{R}_u$ becomes small if there are many short paths between two vertices in the graph [24]. In such cases, removing an edge has smaller impact on connectivity or message propagation due to the availability of alternative paths. The proof is provided in Appendix C.4. The cases for link prediction and graph classification are shown in Appendices B.5 and B.6

**Theorem 4.8** (Effective resistance bounds the robustness of explanations). *For node $u$, let $\bar{R}_s = \frac{\sum_{(q,v) \in \mathcal{E}_s} R_{q,v}}{|\mathcal{E}_s|}$ denote the average effective resistance of edges in $\mathcal{G}_s$, and $d_{\min} = \min_{v \in \mathcal{V}_s} \deg(v)$. Let $\mathcal{G}' = (\mathcal{V}', \mathcal{E}'), \mathcal{E}_s \subseteq \mathcal{E}'$ denote perturbed graph by adding and/or deleting edges not in $\mathcal{E}_s$. Let $\gamma$ denote the maximum eigenvalue of $\hat{A}$. Whether $\bigoplus$ is the sum or mean operation, the following inequality holds for the node classification task,*

$$|\mathrm{Pr}_u(\mathcal{G}) - \mathrm{Pr}_u(\mathcal{G}')| \le L_1 L^T M^T C \left( 2|\mathcal{V}| + \bar{R}_s + \frac{2}{d_{min}} \frac{1}{1 - \gamma} + 2 \right).$$

Theorem 4.8 shows that a smaller $\bar{R}_s$ implies greater robustness of the model explanations under input perturbations. The proof is provided in Appendix C.5. The cases for link prediction and graph classification tasks are shown in Appendices B.7 and B.8

## 5 Method

Existing explanation methods compute the importance score $F(u, v)$ for each edge $(u, v)$ to faithfully explain model predictions. However, $F(u, v)$ fails to capture the sensitivity of edges to perturbations, resulting in potentially non-robust explanations. Thus, we incorporate Ricci curvature $\kappa(u, v)$ and effective resistance $R(u, v)$ as the robust scores of edges. Let $\mathcal{G}_s = (\mathcal{V}_s, \mathcal{E}_s)$ denote the explanation subgraph and $x(u, v)$ denote whether edge $(u, v)$ is in $\mathcal{E}_s$. To balance fidelity and robustness, we define the following objective functions to select important edges,

$$\max_{\substack{x(u,v) \in \{0,1\} \\ \sum_{(u,v) \in \mathcal{E}} x(u,v) = |\mathcal{E}_s|}} x(u, v) \left( F(u, v) + \lambda \mathcal{C}(u, v) \right), \quad \text{where} \quad \mathcal{C}(u, v) = \begin{cases} \kappa(u, v), \\ -R(u, v), \end{cases} \quad (7)$$

$\lambda$ is the parameter to control the trade-off between fidelity and robustness. Due to a negative correlation between Ricci curvature and effective resistance (see Appendix D.5), we construct robust explanations using each metric separately. For curvature-based explanations, we select edges with higher $F(u, v)$ and $\kappa(u, v)$. For resistance-based explanations, we select edges with higher $F(u, v)$ and lower $R(u, v)$. A more robust explanation is obtained using Equation (7).

**Computational complexity.** The complexity of Ollivier-Ricci curvature is $O(|\mathcal{E}| \cdot |\mathcal{V}|^3)$. Computing effective resistance requires the pseudoinverse of $\hat{L}$, which is computationally expensive. To address this, we adopt an efficient approximation method based on solving linear systems of the form $Lx = b$ [43], with a complexity of $O(|\mathcal{E}| \cdot \log(|\mathcal{V}|))$.

# 6  Experiments

**Datasets and tasks**. We evaluate our method on three tasks across nine datasets: Cora, Citeseer, and PubMed for node classification; BC-OTC, BC-Alpha, and UCI for link prediction; and MUTAG, PROTEINS, and IMDB-BINARY for graph classification. Details are provided in Appendix D.1.

**Explanation methods**. We consider six explanation methods: GNNExplainer, PGExplainer, GNN-LRP, DeepLIFT, FlowX, and Convex. Details of these methods are provided in Appendix D.4.

**Experimental setup**. For each dataset, we train a GNN on the training set according to the task. Two GNN architectures are evaluated, with implementation details provided in Appendix D.2. We apply six classical explanation methods to compute edge importance scores $F(u, v)$ and calculate Ricci curvature and effective resistance for all edges. To determine the optimal $\lambda$ in Equation (7), we split the explanation targets into training and test subsets. Optuna is used to tune $\lambda$ on the training subset, which is then fixed on the test subset used in Equation (7). The selected $\lambda$ values for Ricci curvature and effective resistance across methods and datasets are shown in Tables 11, 12, 13 and 14.

Table 1: Relative error performance for robust explanations based on **Ricci curvature**. The $\bigoplus$ in GNN is **sum** operation. A higher value ($\uparrow$) indicates better performance. Blue highlights the best result.

| Methods | Cora | Citeseer | Pubmed | BC-OTC | BC-Alpha | UCI | MUTAG | PROTEINS | IMDB-BINARY |
|---|---|---|---|---|---|---|---|---|---|
| **GNNExplainer** | 4.5% | -6.0% | 0.5% | 54.4% | 70.1% | 73.3% | 2.2% | 4.0% | -0.2% |
| **GNNExplainer+Curvature** | 16.0% | 2.4% | 2.6% | 55.2% | 70.5% | 74.7% | 5.1% | 5.1% | 1.5% |
| **PGExplainer** | 2.7% | 6.1% | 2.6% | 3.8% | -0.3% | -0.8% | -2.8% | 0.2% | -1.2% |
| **PGExplainer+Curvature** | 4.5% | 8.4% | 5.6% | 6.4% | 1.9% | 1.9% | 3.9% | 2.6% | 1.3% |
| **Convex** | 8.4% | 41.8% | 32.5% | 35.3% | 60.2% | 47.1% | 0.4% | -4.3% | 8.1% |
| **Convex+Curvature** | 15.9% | 60.6% | 37.7% | 56.3% | 71.2% | 58.2% | 5.6% | 1.1% | 12.3% |
| **DeepLIFT** | 11.6% | 15.8% | 13.4% | 0.3% | -2.1% | 11.1% | -2.5% | 0.5% | -0.5% |
| **DeepLIFT+Curvature** | 16.7% | 23.9% | 14.6% | 1.3% | 1.9% | 11.5% | 1.4% | 2.1% | 1.9 |
| **GNNLRP** | 16.4% | -2.0% | 19.4% | 7.5% | 5.1% | 19.8% | -1.1% | 0% | -2.7% |
| **GNNLRP+Curvature** | 22.7% | 6.2% | 23.3% | 9.1% | 5.7% | 24.4% | 2.5% | 1.5% | 2.7% |
| **FlowX** | 2.7% | 30.6% | 8.6% | 18.3 % | 3.5% | 7.0% | -1.8% | -3.1% | 0.5% |
| **FlowX+Curvature** | 7.8% | 38.3% | 17.5% | 19.3% | 4.3% | 11.9% | 1.3% | 1.6% | 1.8% |

Table 2: Fidelity$_{KL}$ performance for robust explanations based on **Ricci curvature.** The $\bigoplus$ in GNN is **sum** operation. A lower value ($\downarrow$) indicates better performance, with blue highlighting the best result.

| Methods | Cora | Citeseer | Pubmed | BC-OTC | BC-Alpha | UCI | MUTAG | PROTEINS | IMDB-BINARY |
|---|---|---|---|---|---|---|---|---|---|
| **GNNExplainer** | 0.961 | 0.578 | 0.549 | 0.184 | 0.245 | 0.093 | 0.177 | 0.068 | 0.284 |
| **GNNExplainer+Curvature** | 0.924 | 0.518 | 0.566 | 0.175 | 0.242 | 0.093 | 0.170 | 0.067 | 0.282 |
| **PGExplainer** | 1.762 | 1.341 | 0.715 | 0.537 | 0.447 | 0.400 | 0.227 | 0.048 | 0.351 |
| **PGExplainer+Curvature** | 1.754 | 1.339 | 0.726 | 0.537 | 0.447 | 0.399 | 0.225 | 0.048 | 0.344 |
| **Convex** | 0.353 | 0.277 | 0.075 | 0.132 | 0.009 | 0.011 | 0.010 | 0.001 | 0.017 |
| **Convex+Curvature** | 0.340 | 0.273 | 0.073 | 0.053 | 0.008 | 0.010 | 0.008 | 0.001 | 0.021 |
| **DeepLIFT** | 1.020 | 0.481 | 0.277 | 0.178 | 0.134 | 0.202 | 0.206 | 0.054 | 0.315 |
| **DeepLIFT+Curvature** | 0.956 | 0.473 | 0.304 | 0.177 | 0.133 | 0.202 | 0.180 | 0.054 | 0.321 |
| **GNNLRP** | 0.472 | 0.589 | 0.293 | 0.174 | 0.202 | 0.234 | 0.116 | 0.033 | 0.303 |
| **GNNLRP+Curvature** | 0.442 | 0.572 | 0.276 | 0.173 | 0.201 | 0.234 | 0.153 | 0.036 | 0.321 |
| **FlowX** | 0.495 | 0.499 | 0.370 | 0.184 | 0.086 | 0.244 | 0.174 | 0.051 | 0.298 |
| **FlowX+Curvature** | 0.489 | 0.470 | 0.365 | 0.182 | 0.085 | 0.244 | 0.171 | 0.051 | 0.301 |

**Quantitative evaluation metrics**. We adopt the robustness metric $\delta^*$ proposed in [40], defined in Equation (2), where lower values indicate more robust explanations. However, in some cases, $\delta^*$ is small and cannot significantly show difference across methods. We report the relative error, defined as $\frac{\delta^*_{\text{ranodm}} - \delta^*}{\delta^*_{\text{ranodm}}}$, where $\delta^*_{\text{ranodm}}$ is the robustness of a randomly sampled explanation subgraph. A negative relative error indicates worse robustness than the random baseline, while a higher value implies stronger robustness by reflecting greater improvement over randomness. We also evaluate fidelity, which quantifies how well the explanation subgraph preserves the model's predictive distribution.

Following [15, 29], we use the KL-based metric: $\text{Fidelity}_{KL} = KL(\text{Pr}(\mathcal{G}) \| \text{Pr}(\mathcal{G}_s))$, where lower $\text{Fidelity}_{KL}$ values suggest that $\mathcal{G}_s$ is important for model. More details can be seen in Appendix D.3.

Table 3: Relative error performance for robust explanations based on **effective resistance**. The $\bigoplus$ in GNN is **sum** operation. A higher value ($\uparrow$) indicates better performance. Blue highlights the best result.

| Methods | Cora | Citeseer | Pubmed | BC-OTC | BC-Alpha | UCI | MUTAG | PROTEINS | IMDB-BINARY |
|---|---|---|---|---|---|---|---|---|---|
| **GNNExplainer** | 13.6% | 32.0% | 5.8% | 19.8% | 14.6% | 41.6% | 1.1% | 0.9% | 0.4% |
| **GNNExplainer-Curvature** | 14.4% | 36.5% | 7.7% | 25.6% | 19.2% | 50.7% | 5.3% | 2.3% | 3.1% |
| **PGExplainer** | -1.5% | 0.5% | 0.6% | 1.7% | 3.9% | 11.3% | -3.9% | 1.7% | -1.8% |
| **PGExplainer-Curvature** | 1.0% | 1.5% | 3.2% | 11.6% | 14.2% | 25.4% | 3.5% | 2.2% | 2.0% |
| **Convex** | 40.9% | 32.6% | 21.2% | 63.7% | 56.9% | 33.7% | 1.7% | -3.1% | 4.3% |
| **Convex-Curvature** | 46.0% | 40.5% | 29.1% | 72.7% | 58.3% | 63.7% | 4.7% | 2.1% | 5.6% |
| **DeepLIFT** | 14.9% | 5.5% | 14.7% | -3.4% | 0.1% | -3.8% | -1.1% | 0.9% | -1.0% |
| **DeepLIFT-Curvature** | 17.2% | 19.5% | 18.5% | 0.3% | 1.0% | 1.5% | 3.2% | 1.8% | 0.8% |
| **GNN-LRP** | 57.9% | 24.3% | 43.7% | 2.6 % | 0.2% | 1.9% | 1.7% | -0.9% | -1.2% |
| **GNN-LRP-Curvature** | 60.5% | 25.5% | 45.4% | 6.0% | 1.4% | 2.3% | 3.8% | 0.7% | 1.3% |
| **FlowX** | 14.0% | 12.3% | 21.5% | -2.5% | 7.4% | 3.9% | -2.9% | -0.8% | -0.9% |
| **FlowX-Curvature** | 19.0% | 13.3% | 22.7% | 25.5% | 24.1% | 5.3% | 3.6% | 0.2% | 0.3% |

Table 4: $\text{Fidelity}_{KL}$ performance for robust explanations based on **effective resistance**. The $\bigoplus$ in GNN is **sum** operation. A lower value ($\downarrow$) indicates better performance, with blue highlighting the best result.

| Methods | Cora | Citeseer | Pubmed | BC-OTC | BC-Alpha | UCI | MUTAG | PROTEINS | IMDB-BINARY |
|---|---|---|---|---|---|---|---|---|---|
| **GNNExplainer** | 1.163 | 0.557 | 0.516 | 0.219 | 0.335 | 0.200 | 0.189 | 0.050 | 0.293 |
| **GNNExplainer-Curvature** | 0.983 | 0.546 | 0.485 | 0.212 | 0.325 | 0.145 | 0.169 | 0.049 | 0.276 |
| **PGExplainer** | 1.573 | 1.521 | 0.695 | 0.254 | 0.433 | 0.392 | 0.199 | 0.054 | 0.297 |
| **PGExplainer-Curvature** | 1.506 | 1.262 | 0.690 | 0.246 | 0.391 | 0.404 | 0.190 | 0.052 | 0.258 |
| **Convex** | 0.195 | 0.165 | 0.071 | 0.011 | 0.001 | 0.046 | 0.108 | 0.001 | 0.012 |
| **Convex-Curvature** | 0.159 | 0.113 | 0.060 | 0.006 | 0.001 | 0.017 | 0.143 | 0.001 | 0.011 |
| **DeepLIFT** | 0.877 | 0.435 | 0.357 | 0.316 | 0.328 | 0.439 | 0.272 | 0.044 | 0.318 |
| **DeepLIFT-Curvature** | 0.833 | 0.457 | 0.361 | 0.314 | 0.326 | 0.449 | 0.258 | 0.044 | 0.320 |
| **GNN-LRP** | 0.096 | 0.257 | 0.397 | 0.304 | 0.317 | 0.290 | 0.152 | 0.059 | 0.255 |
| **GNN-LRP-Curvature** | 0.073 | 0.228 | 0.367 | 0.302 | 0.315 | 0.340 | 0.153 | 0.058 | 0.262 |
| **FlowX** | 0.941 | 0.631 | 0.519 | 0.245 | 0.321 | 0.254 | 0.188 | 0.047 | 0.308 |
| **FlowX-Curvature** | 0.869 | 0.632 | 0.518 | 0.202 | 0.357 | 0.269 | 0.178 | 0.046 | 0.307 |

Table 5: Relative error performance for robust explanations based on **Ricci curvature**. The $\bigoplus$ in GNN is **mean** operation. A higher value ($\uparrow$) indicates better performance. Blue highlights the best result.

| Methods | Cora | Citeseer | Pubmed | BC-OTC | BC-Alpha | UCI | MUTAG | PROTEINS | IMDB-BINARY |
|---|---|---|---|---|---|---|---|---|---|
| **GNNExplainer** | -1.1% | 1.1% | 2.0% | 24.9% | 12.2% | 2.1% | -2.5% | -0.3% | 0.2% |
| **GNNExplainer+Curvature** | 4.8% | 6.8% | 5.5% | 27.9% | 13.5% | 5.3% | 1.3% | 0.5% | 0.6% |
| **PGExplainer** | -0.6% | -0.2% | -0.7% | -0.5% | -1.7% | -1.1% | -0.3% | -0.2% | 0.9% |
| **PGExplainer+Curvature** | 2.4% | 2.4% | 1.2% | 1.8% | 0.5% | 0.9% | 0.9% | 0.5% | 0.3% |
| **Convex** | 23.4% | -3.6% | 4.6% | 14.0% | 2.3% | 9.5% | 0.5% | -4.3% | -0.5% |
| **Convex+Curvature** | 37.9% | 10.8% | 11.3% | 16.8% | 3.5% | 16.7% | 3.2% | 1.1% | 0.2% |
| **DeepLIFT** | -0.1% | 55.3% | 14.9% | -0.9% | -2.3% | 0.8% | -0.2% | 0% | -1.8% |
| **DeepLIFT+Curvature** | 1.7% | 67.9% | 19.8% | 0.4% | 0.5% | 1.7% | 0.3% | 2.2% | 1.0% |
| **GNNLRP** | 20.2% | 8.9% | 10.3% | 22.9% | 22.8% | -2.2% | -0.5% | -0.7% | 0.3% |
| **GNNLRP+Curvature** | 22.4% | 13.7% | 14.2% | 23.5% | 26.6% | 3.6% | 1.7% | 0.2% | 0.8% |
| **FlowX** | 7.0% | 6.8% | 6.8% | 0.4% | -0.6% | -1.6% | 0.6 % | -0.2% | 0.1% |
| **FlowX+Curvature** | 7.6% | 7.9% | 8.1% | 1.7% | 0.2% | 0.7% | 1.1% | 0.3% | 1.3% |

**Performance evaluation and comparison**. We report the relative error and fidelity of explanations under different aggregation functions in GNNs. If the $\bigoplus$ is sum operation, the corresponding results are presented in Tables 1, 3 for relative error, and Tables 2, 4 for fidelity. If the $\bigoplus$ is mean operation, results are shown in Tables 5 and 7 for relative error, and Tables 6 and 8 for fidelity. Across all datasets and methods, curvature- and resistance-based explanations consistently improve robustness. In terms

Table 6: Fidelity$_{KL}$ performance for robust explanations based on **Ricci curvature**. The $\bigoplus$ in GNN is **mean** operation. A lower value ($\downarrow$) indicates better performance, with blue highlighting the best result.

| Methods | Cora | Citeseer | Pubmed | BC-OTC | BC-Alpha | UCI | MUTAG | PROTEINS | IMDB-BINARY |
|---|---|---|---|---|---|---|---|---|---|
| **GNNExplainer** | 1.458 | 1.361 | 0.162 | 0.176 | 0.099 | 0.063 | 0.153 | 0.034 | 0.241 |
| **GNNExplainer+Curvature** | 1.393 | 1.372 | 0.149 | 0.175 | 0.098 | 0.063 | 0.150 | 0.032 | 0.235 |
| **PGExplainer** | 1.647 | 1.534 | 0.581 | 0.209 | 0.110 | 0.054 | 0.156 | 0.025 | 0.249 |
| **PGExplainer+Curvature** | 1.664 | 1.483 | 0.582 | 0.206 | 0.109 | 0.053 | 0.146 | 0.024 | 0.243 |
| **Convex** | 0.086 | 0.005 | 0.002 | 0.033 | 0.007 | 0.001 | 0.015 | 0.001 | 0.011 |
| **Convex+Curvature** | 0.029 | 0.002 | 0.002 | 0.030 | 0.004 | 0.001 | 0.019 | 0.001 | 0.011 |
| **DeepLIFT** | 1.012 | 0.291 | 0.270 | 0.193 | 0.089 | 0.050 | 0.143 | 0.040 | 0.302 |
| **DeepLIFT+Curvature** | 0.992 | 0.260 | 0.270 | 0.192 | 0.089 | 0.049 | 0.126 | 0.040 | 0.301 |
| **GNNLRP** | 0.531 | 0.633 | 0.193 | 0.091 | 0.022 | 0.011 | 0.162 | 0.034 | 0.231 |
| **GNNLRP+Curvature** | 0.470 | 0.529 | 0.176 | 0.104 | 0.023 | 0.013 | 0.173 | 0.034 | 0.229 |
| **FlowX** | 1.276 | 1.164 | 0.466 | 0.190 | 0.081 | 0.054 | 0.161 | 0.039 | 0.253 |
| **FlowX+Curvature** | 1.267 | 1.164 | 0.466 | 0.190 | 0.081 | 0.054 | 0.160 | 0.038 | 0.249 |

Table 7: Relative error performance for robust explanations based on **effective resistance**. The $\bigoplus$ in GNN is **mean** operation. A higher value ($\uparrow$) indicates better performance. Blue highlights the best result.

| Methods | Cora | Citeseer | Pubmed | BC-OTC | BC-Alpha | UCI | MUTAG | PROTEINS | IMDB-BINARY |
|---|---|---|---|---|---|---|---|---|---|
| **GNNExplainer** | 0.8% | 0.6% | 1.1% | 22.0% | 7.4% | -0.7% | 0.3% | -0.2% | -0.1% |
| **GNNExplainer-Curvature** | 4.0% | 3.8% | 5.6% | 23.8% | 8.3% | 2.8% | 2.3% | 0.2% | 0.2% |
| **PGExplainer** | -0.6% | 2.3% | 5.0% | -0.8% | -0.4% | 1.1% | -4.3% | 0.1% | -0.2% |
| **PGExplainer-Curvature** | 3.7% | 4.1% | 5.6% | 2.1% | 3.3% | 2.1% | 1.4% | 0.5% | 0.2% |
| **Convex** | 20.6% | 20.5% | 1.1% | 17.8% | 10.7% | -2.6% | 0.7% | -0.9% | 1.3% |
| **Convex-Curvature** | 22.8% | 28.9% | 2.5% | 19.9% | 15.5% | 3.1% | 1.7% | 2.9% | 2.3% |
| **DeepLIFT** | 16.3% | 19.8% | 23.5% | -0.5% | -4.3% | -0.6% | -1.3% | -0.6% | 0.0% |
| **DeepLIFT-Curvature** | 25.9% | 21.5% | 27.6% | 0.8% | 0.6% | 1.5% | 1.2% | 0.3% | 0.4% |
| **GNN-LRP** | 19.9% | 12.8% | 11.8% | 25.8% | 14.1% | 1.1% | 3.7% | -0.3% | 0.3% |
| **GNN-LRP-Curvature** | 25.5% | 18.2% | 17.6% | 30.7% | 15.5% | 2.7% | 4.9% | 0.4% | 0.6% |
| **FlowX** | 7.1% | 0.2% | 6.3% | 1.0% | 0.8% | -1.9% | 0.3% | 0.1% | 0.7% |
| **FlowX-Curvature** | 8.5% | 2.9% | 7.4% | 2.1% | 2.6% | 1.7% | 2.9% | 0.3% | 0.9% |

of fidelity, Ricci- and resistance-based explanations match or exceed the base explanations in **85**% and **80**% of cases, respectively, under the mean aggregator, and in **83**% and **80**% of cases under the sum aggregator. The impact of robustness enhancement on fidelity varies by method. For GNNExplainer, robustness improvements do not clearly reduce fidelity. However, for GNN-LRP and FlowX, fidelity may decline, especially when importance scores are uniform across edges, causing curvature or resistance to dominate selection and shift the explanation away from the model's original behavior. While improving robustness may slightly affect fidelity, the trade-offs are generally acceptable.

Table 8: Fidelity$_{KL}$ performance for robust explanations based on **effective resistance**. The $\bigoplus$ in GNN is **mean** operation. A lower value ($\downarrow$) indicates better performance, with blue highlighting the best result.

| Methods | Cora | Citeseer | Pubmed | BC-OTC | BC-Alpha | UCI | MUTAG | PROTEINS | IMDB-BINARY |
|---|---|---|---|---|---|---|---|---|---|
| **GNNExplainer** | 1.565 | 1.349 | 0.477 | 0.205 | 0.125 | 0.065 | 0.145 | 0.034 | 0.263 |
| **GNNExplainer-Curvature** | 1.422 | 1.263 | 0.446 | 0.203 | 0.123 | 0.065 | 0.135 | 0.033 | 0.262 |
| **PGExplainer** | 1.606 | 1.455 | 0.560 | 0.343 | 0.158 | 0.065 | 0.181 | 0.034 | 0.235 |
| **PGExplainer-Curvature** | 1.602 | 1.452 | 0.558 | 0.343 | 0.157 | 0.065 | 0.157 | 0.034 | 0.233 |
| **Convex** | 0.049 | 0.054 | 0.014 | 0.020 | 0.007 | 0.015 | 0.004 | 0.001 | 0.002 |
| **Convex-Curvature** | 0.052 | 0.051 | 0.011 | 0.017 | 0.006 | 0.016 | 0.004 | 0.001 | 0.002 |
| **DeepLIFT** | 0.825 | 0.846 | 0.387 | 0.199 | 0.084 | 0.055 | 0.240 | 0.035 | 0.227 |
| **DeepLIFT-Curvature** | 0.806 | 0.849 | 0.397 | 0.199 | 0.084 | 0.055 | 0.233 | 0.035 | 0.226 |
| **GNN-LRP** | 0.568 | 0.802 | 0.317 | 0.034 | 0.027 | 0.101 | 0.133 | 0.034 | 0.202 |
| **GNN-LRP-Curvature** | 0.499 | 0.610 | 0.277 | 0.033 | 0.026 | 0.101 | 0.139 | 0.035 | 0.205 |
| **FlowX** | 1.325 | 1.193 | 0.423 | 0.288 | 0.141 | 0.068 | 0.162 | 0.037 | 0.246 |
| **FlowX-Curvature** | 1.298 | 1.178 | 0.413 | 0.296 | 0.143 | 0.068 | 0.157 | 0.037 | 0.249 |

**Ablation analysis**. To evaluate the impact of $\lambda$ on explanation robustness and fidelity, we compute the differences in relative error and fidelity between the curvature-based and original methods across various datasets, methods, GNN architectures, and $\lambda$ values. A positive difference in relative error suggests enhanced robustness, while a negative difference in fidelity implies an improvement in explanation fidelity relative to the original method. The results are in Figures 5, 6, 7, and 8 (in Appendix). Results show that on some datasets, regardless of whether the GNN uses mean or sum aggregation and across different values of $\lambda$, leveraging Ricci curvature and effective resistance yields more robust and faithful explanations.

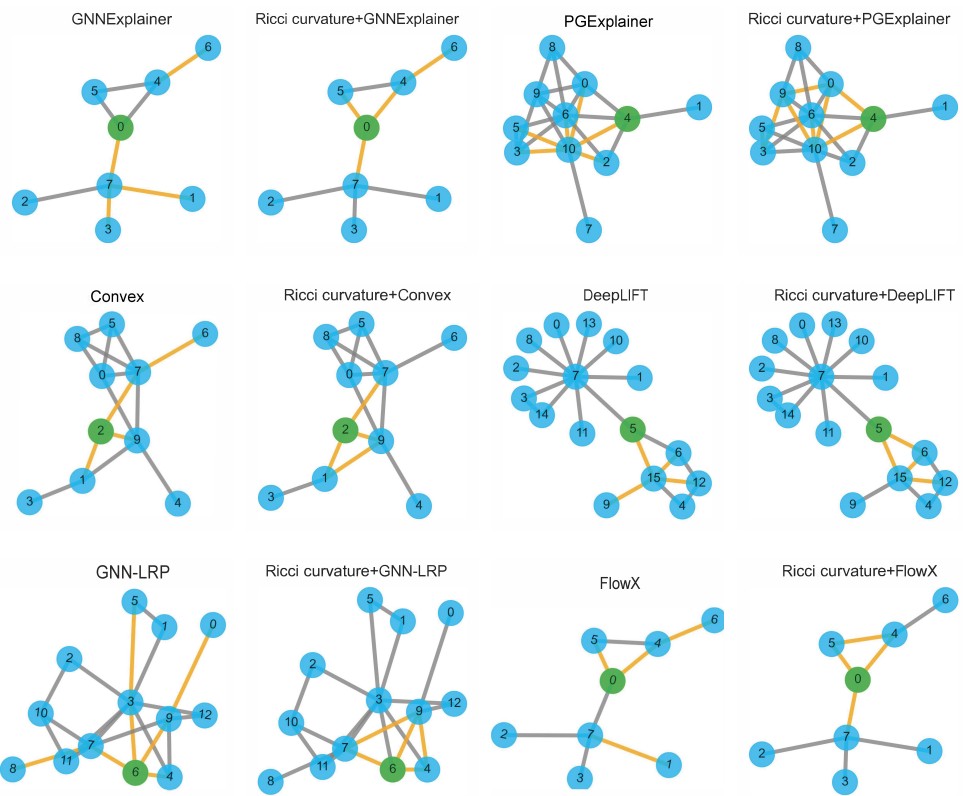

Figure 2: The green node denotes the target node to be explained, and yellow edges indicate the selected important edges. Titles with "Ricci curvature+" indicate that the explanations are curvature-based explanation.

**Running time and case study**. We report the runtime for computing Ricci curvature and effective resistance on three large datasets (details in Table 10). Results are shown in Figures 3 and 4. Ricci curvature takes 3.5 seconds for over 15,000 edges, while effective resistance takes about 80 seconds. These times are acceptable. Visualization results are shown in Figure 2. Explanations considering Ricci curvature tend to select edges forming triangular structures, leading to greater robustness.

## 7 Conclusions and Limitations

We are the first to explore the relationship between the robustness explanations and graph curvatures. We theoretically prove that minimum Ollivier-Ricci curvature and averagr effective resistance can bound the robustness of model predictions and explanations. We use Ricci curvature and effective resistance as robustness scores to obtain roust explanations. Our method is agnostic to specific GNN explanations. Empirically, it consistently improves robustness over six baselines across nine datasets spanning node classification, link prediction, and graph classification, while also improving fidelity in 80% of cases and achieving up to a 10% relative gain in robustness. As a limitation, our analysis focuses the robustness of explanations under structural perturbations, extending it to feature-space is a promising direction for future work.

## Acknowledgements

Hui Xiong was supported in part by the National Key R&D Program of China (Grant No.2023YFF0725001), in part by the National Natural Science Foundation of China (Grant No.92370204), in part by the guangdong Basic and Applied Basic Research Foundation (Grant No.2023B 1515120057), in part by the Education Bureau of Guangzhou. Sihong Xie was supported by the Department of Science and Technology of Guangdong Province (2023CX10X079), National Key R&D Program of China (Grant No.2023YFF0725001), the Guangzhou-HKUST(GZ) Joint Funding Program (Grant No.2023A03J0008), and Education Bureau Guangzhou Municipality. Xi Zhang was supported by the Natural Science Foundation of China (No. 62372057).

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

# Technical Appendices and Supplementary Material

## A The definition of robustness explanations on link prediction and graph classification tasks

**Definition A.1.** Given an input graph $\mathcal{G}$ and explanation $\mathcal{G}_s = (\mathcal{V}_s, \mathcal{E}_s)$, we construct a perturbed graph $\mathcal{G}' = (\mathcal{V}', \mathcal{E}')$ by adding and/or deleting edges not in $\mathcal{E}_s$. The robustness of explanation on graph classification task is defined as:

$$\delta^* = \mathbb{E}_{\mathcal{G}'}|\mathsf{Pr}(\mathcal{G})_c - \mathsf{Pr}(\mathcal{G}')_c| \text{ s.t. } c = \arg\max_i \mathsf{Pr}(\mathcal{G})_i, \mathcal{E}_s \subseteq \mathcal{E}'$$

where $c$ denotes the predicted class of original graph $\mathcal{G}$. $\mathsf{Pr}(\mathcal{G})_i$ and $\mathsf{Pr}(\mathcal{G}')_i$ denote the probability of graph $\mathcal{G}$ and $\mathcal{G}'$ for given class $i$. Lower value indicates greater robustness of the explanation.

**Definition A.2.** Given an input graph $\mathcal{G}$ and explanation $\mathcal{G}_s = (\mathcal{V}_s, \mathcal{E}_s)$, we construct a perturbed graph $\mathcal{G}' = (\mathcal{V}', \mathcal{E}')$ by adding and/or deleting edges not in $\mathcal{E}_s$. For target edge $(u, v)$), the robustness of explanation on link prediction task is defined as:

$$\delta^* = \mathbb{E}_{\mathcal{G}'}|\mathsf{Pr}_{uv}(\mathcal{G})_c - \mathsf{Pr}_{uv}(\mathcal{G}')_c| \text{ s.t. } c = \arg\max_i \mathsf{Pr}_{uv}(\mathcal{G})_i, \mathcal{E}_s \subseteq \mathcal{E}'$$

where $c$ denotes the predicted class of original graph $\mathcal{G}$. $\mathsf{Pr}_{uv}(\mathcal{G})_i$ and $\mathsf{Pr}_{uv}(\mathcal{G}')_i$ denote the probability of edge $(u, v)$ on graph $\mathcal{G}$ and $\mathcal{G}'$ for given class $i$. Lower value indicates greater robustness of the explanation.

## B Theorem

### B.1 The Ricci curvature and model prediction robustness on link prediction task.

**Theorem B.1** (Curvature bounds the robustness of model predictions in link prediction)**.** *Given $\mathcal{G} = (\mathcal{V}, \mathcal{E})$, $\mathcal{G}' = (\mathcal{V}', \mathcal{E}')$ and the $T$ layer model. For $u, v$, let $\mathcal{G}_{u,v} = (\mathcal{V}_{u,v}, \mathcal{E}_{u,v})$ denote the denote the union of $T$-hop subgraphs of $u$ and $v$. Assume that there exists a constant $\alpha' > 0$ such that for edges $(u, v) \in \mathcal{E}_{u,v}$, the curvature is bounded by $\kappa(u, v) \geq \alpha' > 0$. Let $\beta'$ be the maximum degree of nodes in the $T$-hop subgraph of node $u$ for both the original graph $\mathcal{G}$ and the perturbed graph $\mathcal{G}'$. Let $\eta = L_1 L^T M^T C$. Then, the following inequality holds*

**If $\bigoplus$ is the sum operation,**

$$|\mathsf{Pr}_{uv}(\mathcal{G}) - \mathsf{Pr}_{uv}(\mathcal{G}')| \leq \sqrt{2}\eta\left(\sum_{i=2}^{T} 2(\beta')^i (1 + \beta')^{T-i} + (1 + \beta')^{T-1}(2\beta'(1 - \alpha') + 4\beta')\right).$$

**If $\bigoplus$ is the mean operation,**

$$|\mathsf{Pr}(\mathcal{G}) - \mathsf{Pr}(\mathcal{G}')| \leq \sqrt{2}\eta\left(2^{T-1}(4\beta' + \frac{3(1 - \alpha')}{\alpha'}) + 4(T - 1)\beta'\right).$$

### B.2 The Ricci curvature and model prediction robustness on graph classification task.

**Theorem B.2** (Curvature bounds the robustness of model predictions in graph classification)**.** *Given $\mathcal{G} = (\mathcal{V}, \mathcal{E})$, $\mathcal{G}' = (\mathcal{V}', \mathcal{E}')$ and the $T$ layer model. Let $\mathcal{V} = |\mathcal{V}'|$. Assume that there exists a constant $\alpha' > 0$ such that for all edges $(u, v) \in \mathcal{E}$, the curvature is bounded by $\kappa(u, v) \geq \alpha' > 0$. Let $\beta'$ be the maximum degree of nodes in the $T$-hop subgraph of node $u$ for both the original graph $\mathcal{G}$ and the perturbed graph $\mathcal{G}'$. Let $\eta = L_1 L^T M^T C$. Then, the following inequality holds for the graph classification task,*

**If $\bigoplus$ is the sum operation,**

$$|\mathsf{Pr}(\mathcal{G}) - \mathsf{Pr}(\mathcal{G}')| \leq \eta\left(\sum_{i=2}^{T} 2(\beta')^i (1 + \beta')^{T-i} + (1 + \beta')^{T-1}(2\beta'(1 - \alpha') + 4\beta')\right).$$

**If $\bigoplus$ is the mean operation,**

$$|\mathsf{Pr}(\mathcal{G}) - \mathsf{Pr}(\mathcal{G}')| \leq \eta\left(2^{T-1}(4\beta' + \frac{3(1 - \alpha')}{\alpha'}) + 4(T - 1)\beta'\right).$$

*Proof.*

$$|\mathsf{Pr}(\mathcal{G}) - \mathsf{Pr}(\mathcal{G}')| = \left| \frac{\sum_{u \in \mathcal{N}_u(\mathcal{G})} \mathsf{Pr}_u(\mathcal{G})}{|\mathcal{V}|} - \frac{\sum_{u \in \mathcal{N}_u(\mathcal{G}')} \mathsf{Pr}_u(\mathcal{G}')}{|\mathcal{V}|} \right|$$

$$\leq \sum_{u \in \mathcal{N}_u(\mathcal{G})} \frac{|\mathsf{Pr}_u(\mathcal{G}) - \mathsf{Pr}_u(\mathcal{G}')|}{|\mathcal{V}|} \leq \max_{u \in \mathcal{N}_u(\mathcal{G})} |\mathsf{Pr}_u(\mathcal{G}) - \mathsf{Pr}_u(\mathcal{G}')|.$$

According to the Theorem 4.3, and for all edges $(u, v) \in \mathcal{E}$, the curvature is bounded by $\kappa(u, v) \geq \alpha'$, thus, the above inequality holds. $\qquad \square$

### B.3 The Ricci curvature and model explanation robustness on link prediction task.

**Theorem B.3** (Ricci curvature bounds the robustness of explanations on link prediction task). *For edge $(u, v)$, suppose that $\mathcal{G}_s = (\mathcal{V}_s, \mathcal{E}_s)$ is the explanation subgraph of $\mathsf{Pr}_{uv}(\mathcal{G})$. Let $\mathcal{G}' = (\mathcal{V}', \mathcal{E}'), \mathcal{E}_s \subseteq \mathcal{E}'$ denote perturbed graph by adding and/or deleting edges not in $\mathcal{E}_s$. Let $\alpha_s$ denote the minimum $\kappa(u, v)$, such that for edges $(u, v) \in \mathcal{E}_s$, the curvature is bounded by $\kappa(u, v) \geq \alpha_s > 0$. $\eta = L_1 L^T M^T C$. Let $\beta'$ be the maximum degree of nodes in the $T$-hop subgraph of node $u$ for both the original graph $\mathcal{G}$ and the perturbed graph $\mathcal{G}'$, and $\beta_s$ be the maximum degree of nodes in explanation graph $\mathcal{G}_s$. Then,*

*If $\bigoplus$ is the sum operation,*

$$|\mathsf{Pr}_u(\mathcal{G}) - \mathsf{Pr}_u(\mathcal{G}')| \leq \sqrt{2}\eta \left( \left(1 + \beta_s\right)^{T-1} \left(2\beta_s(1 - \alpha_s) + 4\beta_s\right) + \sum_{i=2}^{T} (2\beta'^2 + 2)^i \left(1 + \beta_s\right)^{T-i} \right).$$

*If $\bigoplus$ is the mean operation,*

$$|\mathsf{Pr}_u(\mathcal{G}) - \mathsf{Pr}_u(\mathcal{G}')| \leq \sqrt{2}\eta \left( 2^{T-1} \left(4\beta_s + \frac{3(1 - \alpha_s)}{\alpha_s}\right) + 4(T - 1)\beta' + 2(T - 1) \right).$$

### B.4 The Ricci curvature and model explanation robustness on graph classification task.

**Theorem B.4** (Ricci curvature bounds the robustness of explanations on graph classification task). *Suppose that $\mathcal{G}_s = (\mathcal{V}_s, \mathcal{E}_s)$ is the explanation subgraph of $\mathsf{Pr}(\mathcal{G})$. Let $\mathcal{G}' = (\mathcal{V}', \mathcal{E}'), \mathcal{E}_s \subseteq \mathcal{E}'$ denote perturbed graph by adding and/or deleting edges not in $\mathcal{E}_s$. Let $\alpha_s$ denote the minimum $\kappa(u, v)$, such that for edges $(u, v) \in \mathcal{E}_s$, the curvature is bounded by $\kappa(u, v) \geq \alpha_s > 0$. $\eta = L_1 L^T M^T C$. Let $\beta'$ be the maximum degree of nodes in the $T$-hop subgraph of node $u$ for both the original graph $\mathcal{G}$ and the perturbed graph $\mathcal{G}'$, and $\beta_s$ be the maximum degree of nodes in explanation graph $\mathcal{G}_s$. Then,*

*If $\bigoplus$ is the sum operation,*

$$|\mathsf{Pr}_u(\mathcal{G}) - \mathsf{Pr}_u(\mathcal{G}')| \leq \eta \left( \left(1 + \beta_s\right)^{T-1} \left(2\beta_s(1 - \alpha_s) + 4\beta_s\right) + \sum_{i=2}^{T} (2\beta'^2 + 2)^i \left(1 + \beta_s\right)^{T-i} \right).$$

*If $\bigoplus$ is the mean operation,*

$$|\mathsf{Pr}_u(\mathcal{G}) - \mathsf{Pr}_u(\mathcal{G}')| \leq \eta \left( 2^{T-1} \left(4\beta_s + \frac{3(1 - \alpha_s)}{\alpha_s}\right) + 4(T - 1)\beta' + 2(T - 1) \right).$$

### B.5 The Effective Resistance and model prediction robustness on link prediction task.

**Theorem B.5.** *Given $\mathcal{G} = (\mathcal{V}, \mathcal{E})$ and $\mathcal{G}' = (\mathcal{V}', \mathcal{E}')$ and the $T$ layer model. For $(u, v)$, let $\mathcal{G}_{u,v} = (\mathcal{V}_{u,v}, \mathcal{E}_{u,v})$ denote the union of $T$-hop subgraphs of $u$ and $v$. Let $\bar{R}_u = \frac{\sum_{(q,v) \in \mathcal{E}_{u,v}} R_{q,v}}{|\mathcal{E}_{u,v}|}$ denote the average effective resistance of edges in $\mathcal{G}_{u,v}$. Let $N = |\mathcal{V}_u|$, and $\gamma$ denote the maximum eigenvalue of $\hat{A}$. Let $d_{min} = \min_{v \in \mathcal{V}_{u,v}} \deg(v)$. Let $\eta = L_1 L^T M^T C$. Whether $\bigoplus$ is the mean or sum operation, the following inequality holds for the link prediction task*

$$|\mathsf{Pr}_{uv}(\mathcal{G}) - \mathsf{Pr}_{uv}(\mathcal{G}')| \leq \sqrt{2}\eta \left( 2N + \bar{R}_u + \frac{2}{d_{min}(1 - \gamma)} \right).$$

### B.6 The Effective Resistance and model prediction robustness on graph classification task.

**Theorem B.6.** *Given $\mathcal{G} = (\mathcal{V}, \mathcal{E})$ and $\mathcal{G}' = (\mathcal{V}', \mathcal{E}')$ and the $T$ layer model. Let $\bar{R}_u = \frac{\sum_{(q,v) \in \mathcal{E}_u} R_{q,v}}{|\mathcal{E}|}$ denote the average effective resistance of edges in $\mathcal{G}_u$. Let $N = |\mathcal{V}_u|$, and $\gamma$ denote the maximum eigenvalue of $\hat{A}$. Let $d_{min} = \min_{v \in \mathcal{V}} \deg(v)$. Let $\eta = L_1 L^T M^T C$. Whether $\bigoplus$ is the mean or sum operation, the following inequality holds for the graph classification task*

$$|\text{Pr}(\mathcal{G}) - \text{Pr}(\mathcal{G}')| \leq \eta \left( 2N + \bar{R}_u + \frac{2}{d_{min}(1 - \gamma)} \right).$$

### B.7 The Effective Resistance and explanation robustness on link prediction task.

**Theorem B.7** (Robustness of explanations on link prediction task)**.** *For edge $(u, v)$, suppose that $\mathcal{G}_s = (\mathcal{V}_s, \mathcal{E}_s)$ is the explanation subgraph of $\text{Pr}_{uv}(\mathcal{G})$. Let $\mathcal{G}' = (\mathcal{V}', \mathcal{E}'), \mathcal{E}_s \subseteq \mathcal{E}'$ denote perturbed graph by adding and/or deleting edges not in $\mathcal{E}_s$. Let $\bar{R}_s = \frac{\sum_{(q,v) \in \mathcal{E}_s} R_{q,v}}{|\mathcal{E}_s|}$ denote the average effective resistance of edges in $\mathcal{G}_s$, and $d_{min} = \min_{v \in \mathcal{V}_s} \deg(v)$. Let $\gamma$ denote the maximum eigenvalue of $\hat{A}$. Whether $\bigoplus$ is the sum or mean operation, then,*

$$|\text{Pr}_{uv}(\mathcal{G}_s) - \text{Pr}_{uv}(\mathcal{G}')| \leq \sqrt{2}\eta \left( 2N + \bar{R}_s + \frac{2}{d_{min}(1 - \gamma)} + 2 \right).$$

### B.8 The Effective Resistance and explanation robustness on graph classification task.

**Theorem B.8** (Robustness of explanations on graph classification task)**.** *Suppose that $\mathcal{G}_s = (\mathcal{V}_s, \mathcal{E}_s)$ is the explanation subgraph of $\text{Pr}(\mathcal{G})$. Let $\mathcal{G}' = (\mathcal{V}', \mathcal{E}'), \mathcal{E}_s \subseteq \mathcal{E}'$ denote perturbed graph by adding and/or deleting edges not in $\mathcal{E}_s$. Let $\bar{R}_s = \frac{\sum_{(q,v) \in \mathcal{E}_s} R_{q,v}}{|\mathcal{E}_s|}$ denote the average effective resistance of edges in $\mathcal{G}_s$, and $d_{min} = \min_{v \in \mathcal{V}_s} degree(v)$. Let $\gamma$ denote the maximum eigenvalue of $\hat{A}$. Whether $\bigoplus$ is the sum or mean operation, then,*

$$|\text{Pr}(\mathcal{G}_s) - \text{Pr}(\mathcal{G}')| \leq \eta \left( 2N + \bar{R}_s + \frac{2}{d_{min}(1 - \gamma)} + 2 \right).$$

## C Proofs

### C.1 Proof of Proposition 4.2

*Proof.*
$$\left| \boldsymbol{h}_u^t(\mathcal{G}) - \boldsymbol{h}_u^t(\mathcal{G}') \right| \leq \left| \boldsymbol{h}_u^t(\mathcal{G}) - \boldsymbol{h}_v^t(\mathcal{G}) \right| + \left| \boldsymbol{h}_u^t(\mathcal{G}') - \boldsymbol{h}_v^t(\mathcal{G}) \right| \tag{8}$$

**if $\bigoplus$ is the sum operation**,

$$
\begin{aligned}
\left| \boldsymbol{h}_u^t(\mathcal{G}) - \boldsymbol{h}_v^t(\mathcal{G}) \right| &\leq L \left| \sum_{p \in \mathcal{N}_u} \psi_t(\boldsymbol{h}_p^{t-1}(\mathcal{G})) - \sum_{q \in \mathcal{N}_v} \psi_t(\boldsymbol{h}_q^{t-1}(\mathcal{G})) \right| \\
&= L \left| \sum_{p \in \mathcal{N}_u \backslash \mathcal{N}_v} \psi_t(\boldsymbol{h}_p^{t-1}(\mathcal{G})) - \sum_{q \in \mathcal{N}_v \backslash \mathcal{N}_u} \psi_t(\boldsymbol{h}_q^{t-1}(\mathcal{G})) \right| \\
&\leq L \sum_{p \in \mathcal{N}_u \triangle \mathcal{N}_v} \left| \psi_t(\boldsymbol{h}_p^{t-1}(\mathcal{G})) \right|
\end{aligned}
$$

In [22], the curvature holds the following inequality $\frac{|\mathcal{N}_u \cap \mathcal{N}_v|}{\max(m,n)} \geq \kappa(u, v)$. Then,

$$|\mathcal{N}_v \backslash \mathcal{N}_u| \leq |\mathcal{N}_u \backslash \mathcal{N}_v| = n + 1 - |\mathcal{N}_u \cap \mathcal{N}_v| - 2 \leq n - n\kappa(u, v).$$

The symmetric difference $\mathcal{N}_u \triangle \mathcal{N}_v$ satisfies

$$|\mathcal{N}_u \triangle \mathcal{N}_v| = |(\mathcal{N}_u \backslash \mathcal{N}_v) \cup (\mathcal{N}_v \backslash \mathcal{N}_u)| \leq 2(1 - \kappa(u, v))n. \tag{9}$$

Thus, $|\boldsymbol{h}_u^t(\mathcal{G}) - \boldsymbol{h}_v^t(\mathcal{G})| \leq 2n(1 - \kappa(u,v))LCM$. The second part of Equation (8) is

$$
\left| \boldsymbol{h}_u^t(\mathcal{G}') - \boldsymbol{h}_v^t(\mathcal{G}) \right|
$$

$$
\leq L \left| \sum_{p \in \mathcal{N}_u(\mathcal{G}')} \psi_{t-1}\left( \boldsymbol{h}_p^{t-1}(\mathcal{G}') \right) - \sum_{q \in \mathcal{N}_v(\mathcal{G})} \psi_{t-1}\left( \boldsymbol{h}_q^{t-1}(\mathcal{G}) \right) \right|
$$

$$
\leq L \left| \sum_{p \in \mathcal{N}_u(\mathcal{G}') \backslash \mathcal{N}_v(\mathcal{G})} \psi_{t-1}\left( \boldsymbol{h}_p^{t-1}(\mathcal{G}') \right) \right| + \left| \sum_{q \in \mathcal{N}_v(\mathcal{G}) \backslash \mathcal{N}_u(\mathcal{G}')} \psi_{t-1}\left( \boldsymbol{h}_q^{t-1}(\mathcal{G}) \right) \right|
$$

$$
+ \left| \sum_{p \in \mathcal{N}_u(\mathcal{G}') \cap \mathcal{N}_v(\mathcal{G})} \left( \psi_{t-1}\left( \boldsymbol{h}_p^{t-1}(\mathcal{G}') \right) - \psi_{t-1}\left( \boldsymbol{h}_p^{t-1}(\mathcal{G}) \right) \right) \right|
$$

$$
\leq |\mathcal{N}_u(\mathcal{G}') \,\triangle\, \mathcal{N}_v(\mathcal{G})| \, LCM + 2n'LCM
$$

$$
\leq 4n'LCM
$$

$\beta = max(n, n')$, thus we can obtain the following equation:

$$
\left| \boldsymbol{h}_u^t(\mathcal{G}') - \boldsymbol{h}_u^t(\mathcal{G}) \right| \leq 2n\big(1 - \kappa(u,v)\big)LCM + 4n'LCM \leq 2\beta\big(1 - \alpha\big)LCM + 4\beta LCM \quad (10)
$$

if $\bigoplus$ **is the mean operation**, in [22], we can obtain the following equation:

$$
\left| \boldsymbol{h}_u^t(\mathcal{G}) - \boldsymbol{h}_v^t(\mathcal{G}) \right| \leq L \left| \sum_{p \in \mathcal{N}_u} \frac{1}{n} \psi_t(\boldsymbol{h}_p^{t-1}(\mathcal{G})) - \sum_{q \in \mathcal{N}_v} \frac{1}{m} \psi_t(\boldsymbol{h}_q^{t-1}(\mathcal{G})) \right|
$$

$$
= L \sum_{p \in \mathcal{N}_u \cap \mathcal{N}_v} \left( \frac{1}{m} - \frac{1}{n} \right) \left| \psi_t(\boldsymbol{h}_p^{t-1}(\mathcal{G})) \right|
$$

$$
+ L \left| \sum_{p \in \mathcal{N}_u \backslash \mathcal{N}_v} \frac{1}{n} \psi_t(\boldsymbol{h}_p^{t-1}(\mathcal{G})) - \sum_{q \in \mathcal{N}_v \backslash \mathcal{N}_u} \frac{1}{m} \psi_t(\boldsymbol{h}_q^{t-1}(\mathcal{G})) \right|
$$

We have $n, m \geq |\mathcal{N}_u \cap \mathcal{N}_v| \geq \kappa(u,v)n$.

$$
\frac{1}{m} - \frac{1}{n} \leq \frac{1}{\kappa(u,v)n} - \frac{1}{n} = \frac{1 - \kappa(u,v)}{\kappa(u,v)n}
$$

Then,

$$
\left| \boldsymbol{h}_u^{t+1} - \boldsymbol{h}_v^{t+1} \right| \leq L \sum_{p \in \mathcal{N}_u \cap \mathcal{N}_v} \frac{1 - \kappa(u,v)}{\kappa(u,v)n} \left| \psi_t(\boldsymbol{h}_p^t) \right| + L \sum_{p \in \mathcal{N}_u \triangle \mathcal{N}_v} \frac{1}{\kappa(u,v)n} \left| \psi_t(\boldsymbol{h}_p^t) \right|
$$

$$
\leq Ln \frac{1 - \kappa(u,v)}{\kappa(u,v)n} CM + 2(1 - \kappa(u,v))nL \frac{1}{\kappa(u,v)n} CM
$$

$$
\leq (1 - \kappa(u,v))LCM \left( \frac{3}{\kappa(u,v)} \right)
$$

The second part of Equation (8) can be rewritten as

$$\left|\boldsymbol{h}_u^t(\mathcal{G}') - \boldsymbol{h}_v^t(\mathcal{G})\right|$$

$$\leq L\left|\sum_{p\in\mathcal{N}_u(\mathcal{G}')}\frac{1}{n'}\psi_{t-1}\left(\boldsymbol{h}_p^{t-1}(\mathcal{G}')\right) - \sum_{q\in\mathcal{N}_v(\mathcal{G})}\frac{1}{m}\psi_{t-1}\left(\boldsymbol{h}_q^{t-1}(\mathcal{G})\right)\right|$$

$$\leq L\left|\sum_{p\in\mathcal{N}_u(\mathcal{G}')\backslash\mathcal{N}_v(\mathcal{G})}\frac{1}{n'}\psi_{t-1}\left(\boldsymbol{h}_p^{t-1}(\mathcal{G}')\right)\right| + \left|\sum_{q\in\mathcal{N}_v(\mathcal{G})\backslash\mathcal{N}_u(\mathcal{G}')}\frac{1}{m}\psi_{t-1}\left(\boldsymbol{h}_q^{t-1}(\mathcal{G})\right)\right|$$

$$+ \left|\sum_{p\in\mathcal{N}_u(\mathcal{G}')\cap\mathcal{N}_v(\mathcal{G})}\left(\frac{1}{n'}\psi_{t-1}\left(\boldsymbol{h}_p^{t-1}(\mathcal{G}')\right) - \frac{1}{m}\psi_{t-1}\left(\boldsymbol{h}_p^{t-1}(\mathcal{G})\right)\right)\right|$$

$$\leq \frac{\sum_{p\in\mathcal{N}_u(\mathcal{G}')\backslash\mathcal{N}_v(\mathcal{G})}}{n'}LCM + \frac{\sum_{q\in\mathcal{N}_v(\mathcal{G})\backslash\mathcal{N}_u(\mathcal{G}')}}{m}LCM + 2LCM$$

$$\leq \frac{|\mathcal{N}_u(\mathcal{G}') \vartriangle \mathcal{N}_v(\mathcal{G})|}{m}LCM + 2LCM$$

$$\leq 4LCM$$

Thus,

$$\left|\boldsymbol{h}_u^t(\mathcal{G}') - \boldsymbol{h}_u^t(\mathcal{G})\right| \leq (1 - \kappa(u,v))LCM\left(\frac{3}{\kappa(u,v)}\right) + 4LCM \tag{11}$$

$$\leq \frac{(1-\alpha)3LCM}{\alpha} + 4\beta LCM$$

$$\leq LCM\beta(4 + g_1(\alpha)) \tag{12}$$

$\square$

## C.2  Proof of Theorem 4.3

*Proof.* **If $\bigoplus$ is the sum operation**, and suppose that the GNN has two layer, then,

$$|\mathsf{Pr}_u(\mathcal{G}) - \mathsf{Pr}_u(\mathcal{G}')| \leq L_1\left|\boldsymbol{h}_u^2(\mathcal{G}) - \boldsymbol{h}_u^2(\mathcal{G}')\right|$$

$$\leq L_1 LM\left|\boldsymbol{h}_u^1(\mathcal{G}) - \boldsymbol{h}_u^1(\mathcal{G}') + \sum_{v\in\mathcal{N}_u(\mathcal{G})}\boldsymbol{h}_v^1(\mathcal{G}) - \sum_{v\in\mathcal{N}_u(\mathcal{G}')}\boldsymbol{h}_v^1(\mathcal{G}')\right|$$

$$\leq L_1 LM\left(\left|\boldsymbol{h}_u^1(\mathcal{G}) - \boldsymbol{h}_u^1(\mathcal{G}')\right| + \left|\sum_{v\in\mathcal{N}_u(\mathcal{G})\backslash\mathcal{N}_u(\mathcal{G}')}\boldsymbol{h}_v^1(\mathcal{G})\right| + \left|\sum_{v\in\mathcal{N}_u(\mathcal{G}')\backslash\mathcal{N}_u(\mathcal{G})}\boldsymbol{h}_v^1(\mathcal{G}')\right|\right.$$

$$\left. + \left|\sum_{v\in\mathcal{N}_u(\mathcal{G}')\cap\mathcal{N}_u(\mathcal{G})}\left(\boldsymbol{h}_v^1(\mathcal{G}) - \boldsymbol{h}_v^1(\mathcal{G}')\right)\right|\right)$$

$$\leq L_1 LM\left(\left|\boldsymbol{h}_u^1(\mathcal{G}) - \boldsymbol{h}_u^1(\mathcal{G}')\right| + \left|\sum_{v\in\mathcal{N}_u(\mathcal{G})\backslash\mathcal{N}_u(\mathcal{G}')}\sum_{p\in\mathcal{N}_v(\mathcal{G})}\boldsymbol{h}_p^1(\mathcal{G})\right|\right.$$

$$\left. + \left|\sum_{v\in\mathcal{N}_u(\mathcal{G}')\backslash\mathcal{N}_u(\mathcal{G})}\sum_{p\in\mathcal{N}_v(\mathcal{G}')}\boldsymbol{h}_p^1(\mathcal{G}')\right| + \left|\sum_{v\in\mathcal{N}_u(\mathcal{G}')\cap\mathcal{N}_u(\mathcal{G})}\left(\boldsymbol{h}_v^1(\mathcal{G}) - \boldsymbol{h}_v^1(\mathcal{G}')\right)\right|\right)$$

$\alpha'$ is the minimum curvature of all edges $(u,v) \in \mathcal{E}_u$, and $\beta'$ is the maximum degree of nodes in the $T$-hop subgraph of node $u$ for both the original graph $\mathcal{G}$ and the perturbed graph $\mathcal{G}'$. According to

Equation (10),

$$|\text{Pr}_u(\mathcal{G}) - \text{Pr}_u(\mathcal{G}')|$$

$$\leq L_1 L^2 M^2 C \left( 2\beta'(1-\alpha') + 4\beta' + \beta'^2 + \beta'^2 + \beta' \left( 2\beta'(1-\alpha') + 4\beta' \right) \right)$$

$$\leq L_1 L^2 M^2 C \left( (1+\beta') \left( 2\beta'(1-\alpha') + 4\beta' \right) + 2\beta'^2 \right)$$

Similarly, If the GNN has the $T$ layer, then,

$$|\text{Pr}_u(\mathcal{G}) - \text{Pr}_u(\mathcal{G}')| \leq L_1 \left| \boldsymbol{h}_u^T(\mathcal{G}) - \boldsymbol{h}_u^T(\mathcal{G}') \right|$$

$$\leq L_1 L M \left( \left| \boldsymbol{h}_u^{T-1}(\mathcal{G}) - \boldsymbol{h}_u^{T-1}(\mathcal{G}') \right| + \left| \sum_{v \in \mathcal{N}_u(\mathcal{G}) \setminus \mathcal{N}_u(\mathcal{G}')} \boldsymbol{h}_v^{T-1}(\mathcal{G}) - \sum_{v \in \mathcal{N}_u(\mathcal{G}') \setminus \mathcal{N}_u(\mathcal{G})} \boldsymbol{h}_v^{T-1}(\mathcal{G}') \right| \right.$$

$$\left. + \left| \sum_{v \in \mathcal{N}_u(\mathcal{G}') \cap \mathcal{N}_u(\mathcal{G})} \left( \boldsymbol{h}_v^{T-1}(\mathcal{G}) - \boldsymbol{h}_v^{T-1}(\mathcal{G}') \right) \right| \right)$$

$$\leq \cdots \cdots$$

$$\leq L_1 L^T M^T C \left( (1+\beta')^{T-1} \left( 2\beta'(1-\alpha') + 4\beta' \right) + \sum_{i=2}^{T} 2\beta'^i (1+\beta')^{T-i} \right)$$

**If $\bigoplus$ is the mean operation**, and suppose that the GNN has two layer, and $n' \geq n \geq m$, then,

$$|\text{Pr}_u(\mathcal{G}) - \text{Pr}_u(\mathcal{G}')| \leq L_1 \left| \boldsymbol{h}_u^2(\mathcal{G}) - \boldsymbol{h}_u^2(\mathcal{G}') \right|$$

$$\leq L_1 L M \left| \frac{1}{n} \boldsymbol{h}_u^1(\mathcal{G}) - \frac{1}{n'} \boldsymbol{h}_u^1(\mathcal{G}') + \sum_{v \in \mathcal{N}_u(\mathcal{G})} \frac{1}{n} \boldsymbol{h}_v^1(\mathcal{G}) - \sum_{v \in \mathcal{N}_u(\mathcal{G}')} \frac{1}{n'} \boldsymbol{h}_v^1(\mathcal{G}') \right|$$

$$\leq L_1 L M \left( \left| \frac{1}{n} \boldsymbol{h}_u^1(\mathcal{G}) - \frac{1}{n} \boldsymbol{h}_u^1(\mathcal{G}') + (\frac{1}{n} - \frac{1}{n'}) \boldsymbol{h}_u^1(\mathcal{G}') \right| + \left| \sum_{v \in \mathcal{N}_u(\mathcal{G}) \setminus \mathcal{N}_u(\mathcal{G}')} \frac{1}{n} \boldsymbol{h}_v^1(\mathcal{G}) \right| \right.$$

$$\left. + \left| \sum_{v \in \mathcal{N}_u(\mathcal{G}') \setminus \mathcal{N}_u(\mathcal{G})} \frac{1}{n'} \boldsymbol{h}_v^1(\mathcal{G}') \right| + \left| \sum_{v \in \mathcal{N}_u(\mathcal{G}') \cap \mathcal{N}_u(\mathcal{G})} \left( \frac{1}{n} \boldsymbol{h}_v^1(\mathcal{G}) - \frac{1}{n} \boldsymbol{h}_v^1(\mathcal{G}') + (\frac{1}{n} - \frac{1}{n'}) \boldsymbol{h}_v^1(\mathcal{G}') \right) \right| \right)$$

$$\leq L_1 L M \left( \left| \frac{1}{n} (\boldsymbol{h}_u^1(\mathcal{G}) - \boldsymbol{h}_u^1(\mathcal{G}')) \right| + \left| (\frac{1}{n} - \frac{1}{n'}) \boldsymbol{h}_u^1(\mathcal{G}') \right| + \left| \sum_{v \in \mathcal{N}_u(\mathcal{G}) \setminus \mathcal{N}_u(\mathcal{G}')} \frac{1}{n} \sum_{p \in \mathcal{N}_v(\mathcal{G})} \frac{1}{m} \boldsymbol{h}_p^0(\mathcal{G}) \right| \right.$$

$$+ \left| \sum_{v \in \mathcal{N}_u(\mathcal{G}') \setminus \mathcal{N}_u(\mathcal{G})} \frac{1}{n'} \sum_{p \in \mathcal{N}_v(\mathcal{G}')} \frac{1}{m'} \boldsymbol{h}_p^0(\mathcal{G}') \right|$$

$$\left. + \left| \sum_{v \in \mathcal{N}_u(\mathcal{G}') \cap \mathcal{N}_u(\mathcal{G})} \frac{1}{n} \left( \boldsymbol{h}_v^1(\mathcal{G}) - \boldsymbol{h}_v^1(\mathcal{G}') \right) + (\frac{1}{n} - \frac{1}{n'}) \boldsymbol{h}_v^1(\mathcal{G}') \right| \right)$$

Let $\alpha'$ be the minimum curvature of all edges $(u, v) \in \mathcal{V}_u$, and $\beta'$ be the maximum degree of nodes in the $T$-hop subgraph of node $u$ for both the original graph $\mathcal{G}$ and the perturbed graph $\mathcal{G}'$. According to Equation (12)

$$|\text{Pr}_u(\mathcal{G}) - \text{Pr}_u(\mathcal{G}')|$$

$$\leq L_1 L^2 M^2 C \left( 4\beta' + \frac{3(1-\alpha')}{\alpha'} + \beta' + 2\beta' + 2\beta' + \frac{3(1-\alpha')}{\alpha'} + \beta' \right)$$

$$\leq L_1 L^2 M^2 C \left( 2 \left( 4\beta' + \frac{3(1-\alpha')}{\alpha'} \right) + 4\beta' \right)$$

Similarly, If the GNN has the $T$ layer, then,

$$|\mathrm{Pr}_u(\mathcal{G}) - \mathrm{Pr}_u(\mathcal{G}')| \leq L_1 \left| \boldsymbol{h}_u^T(\mathcal{G}) - \boldsymbol{h}_u^T(\mathcal{G}') \right|$$

$$\leq \cdots \cdots$$

$$\leq L_1 L^T M^T C \left( 2^{T-1} \left( 4\beta' + \frac{3(1-\alpha')}{\alpha'} \right) + 4(T-1)\beta' \right)$$

$\square$

## C.3 Proof of Theorem 4.4

*Proof.* $\mathcal{G}_s = (\mathcal{V}_s, \mathcal{E}_s)$ is the explanation graph. $\mathcal{G}' = (\mathcal{V}', \mathcal{E}')$ is the perturbed graph, where $\mathcal{E}_s \subseteq \mathcal{E}'$, obtained by adding or removing edges not in $\mathcal{E}_s$. Then,

$$|\mathrm{Pr}_u(\mathcal{G}) - \mathrm{Pr}_u(\mathcal{G}')|$$
$$\leq |\mathrm{Pr}_u(\mathcal{G}) - \mathrm{Pr}_u(\mathcal{G}_s)| + |\mathrm{Pr}_u(\mathcal{G}_s) - \mathrm{Pr}_u(\mathcal{G}')|$$

**if $\bigoplus$ is the sum operation**, suppose that the GNN has two layer, and let $\alpha_s$ be the minimum curvature of all edges $(u, v) \in \mathcal{E}_s$, and $\beta'$ is the maximum degree of nodes in the $T$-hop subgraph of node $u$ for both the original graph $\mathcal{G}$ and the perturbed graph $\mathcal{G}'$. $\beta_s$ is the maximum degree of nodes in explanation graph $\mathcal{G}_s$. Then,

$$|\mathrm{Pr}_u(\mathcal{G}_s) - \mathrm{Pr}_u(\mathcal{G}')| \leq L_1 \left| \boldsymbol{h}_u^2(\mathcal{G}_s) - \boldsymbol{h}_u^2(\mathcal{G}') \right|$$

$$\leq L_1 L M \left| \boldsymbol{h}_u^1(\mathcal{G}_s) - \boldsymbol{h}_u^1(\mathcal{G}') + \sum_{v \in \mathcal{N}_u(\mathcal{G}_s)} \boldsymbol{h}_v^1(\mathcal{G}_s) - \sum_{v \in \mathcal{N}_u(\mathcal{G}')} \boldsymbol{h}_v^1(\mathcal{G}') \right|$$

$$\leq L_1 L M \left( \left| \boldsymbol{h}_u^1(\mathcal{G}_s) - \boldsymbol{h}_u^1(\mathcal{G}') \right| + \left| \sum_{v \in \mathcal{N}_u(\mathcal{G}_s) \backslash \mathcal{N}_u(\mathcal{G}')} \boldsymbol{h}_v^1(\mathcal{G}_s) \right| + \left| \sum_{v \in \mathcal{N}_u(\mathcal{G}') \backslash \mathcal{N}_u(\mathcal{G}_s)} \boldsymbol{h}_v^1(\mathcal{G}') \right| \right.$$

$$\left. + \left| \sum_{v \in \mathcal{N}_u(\mathcal{G}') \cap \mathcal{N}_u(\mathcal{G}_s)} \left( \boldsymbol{h}_v^1(\mathcal{G}_s) - \boldsymbol{h}_v^1(\mathcal{G}') \right) \right| \right)$$

$$\leq L_1 L M \left( \left| \boldsymbol{h}_u^1(\mathcal{G}_s) - \boldsymbol{h}_u^1(\mathcal{G}') \right| + \left| \sum_{v \in \mathcal{N}_u(\mathcal{G}_s) \backslash \mathcal{N}_u(\mathcal{G}')} \sum_{p \in \mathcal{N}_v(\mathcal{G}_s)} \boldsymbol{h}_p^1(\mathcal{G}_s) \right| \right.$$

$$\left. + \left| \sum_{v \in \mathcal{N}_u(\mathcal{G}') \backslash \mathcal{N}_u(\mathcal{G}_s)} \sum_{p \in \mathcal{N}_v(\mathcal{G}')} \boldsymbol{h}_p^1(\mathcal{G}') \right| + \left| \sum_{v \in \mathcal{N}_u(\mathcal{G}') \cap \mathcal{N}_u(\mathcal{G}_s)} \left( \boldsymbol{h}_v^1(\mathcal{G}_s) - \boldsymbol{h}_v^1(\mathcal{G}') \right) \right| \right)$$

$$\leq L_1 L^2 M^2 C \left( 2\beta_s(1 - \alpha_s) + 4\beta_s + \beta_s \beta' + \beta'^2 + \beta_s \left( 2\beta_s(1 - \alpha_s) + 4\beta_s \right) \right)$$

$$\leq L_1 L^2 M^2 C \left( (1 + \beta_s) \left( 2\beta_s(1 - \alpha_s) + 4\beta_s \right) + \beta_s \beta' + \beta'^2 \right)$$

$$\leq L_1 L^2 M^2 C \left( (1 + \beta_s) \left( 2\beta_s(1 - \alpha_s) + 4\beta_s \right) + 2\beta'^2 \right)$$

Thus,

$$|\mathrm{Pr}_u(\mathcal{G}) - \mathrm{Pr}_u(\mathcal{G}')|$$
$$\leq |\mathrm{Pr}_u(\mathcal{G}) - \mathrm{Pr}_u(\mathcal{G}_s)| + |\mathrm{Pr}_u(\mathcal{G}_s) - \mathrm{Pr}_u(\mathcal{G}')|$$

$$\leq L_1 L^2 M^2 C \left( (1 + \beta_s) \left( 2\beta_s(1 - \alpha_s) + 4\beta_s \right) + 2\beta'^2 + 2 \right)$$

Similarly, If the GNN has the $T$ layer, then,

$$|\mathsf{Pr}_u(\mathcal{G}) - \mathsf{Pr}_u(\mathcal{G}')|$$

$$\leq L_1 L^T M^T C\left(\left(1+\beta_s\right)^{T-1}\left(2\beta_s(1-\alpha_s)+4\beta_s\right) + \sum_{i=2}^{T}(2\beta'^2+2)^i\left(1+\beta_s\right)^{T-i}\right)$$

**If $\bigoplus$ is the mean operation**, and suppose that the GNN has two layer, $n' \geq n \geq m$, and let $\alpha_s$ be the minimum curvature of all edges $(u,v) \in \mathcal{E}_s$, and $\beta'$ is the maximum degree of nodes in the $T$-hop subgraph of node $u$ for both the original graph $\mathcal{G}$ and the perturbed graph $\mathcal{G}'$. $\beta_s$ is the maximum degree of nodes in explanation graph $\mathcal{G}_s$. Then,

$$|\mathsf{Pr}_u(\mathcal{G}_s) - \mathsf{Pr}_u(\mathcal{G}')| \leq L_1 \left|\boldsymbol{h}_u^2(\mathcal{G}_s) - \boldsymbol{h}_u^2(\mathcal{G}')\right|$$

$$\leq L_1 LM \left|\frac{1}{n}\boldsymbol{h}_u^1(\mathcal{G}_s) - \frac{1}{n'}\boldsymbol{h}_u^1(\mathcal{G}') + \sum_{v\in\mathcal{N}_u(\mathcal{G}_s)}\frac{1}{n}\boldsymbol{h}_v^1(\mathcal{G}_s) - \sum_{v\in\mathcal{N}_u(\mathcal{G}')}\frac{1}{n'}\boldsymbol{h}_v^1(\mathcal{G}')\right|$$

$$\leq L_1 LM \left(\left|\frac{1}{n}\boldsymbol{h}_u^1(\mathcal{G}_s) - \frac{1}{n}\boldsymbol{h}_u^1(\mathcal{G}') + (\frac{1}{n}-\frac{1}{n'})\boldsymbol{h}_u^1(\mathcal{G}')\right| + \left|\sum_{v\in\mathcal{N}_u(\mathcal{G}_s)\backslash\mathcal{N}_u(\mathcal{G}')}\frac{1}{n}\boldsymbol{h}_v^1(\mathcal{G}_s)\right|\right.$$

$$+ \left|\sum_{v\in\mathcal{N}_u(\mathcal{G}')\backslash\mathcal{N}_u(\mathcal{G}_s)}\frac{1}{n'}\boldsymbol{h}_v^1(\mathcal{G}')\right| + \left.\left|\sum_{v\in\mathcal{N}_u(\mathcal{G}')\cap\mathcal{N}_u(\mathcal{G}_s)}(\frac{1}{n}\boldsymbol{h}_v^1(\mathcal{G}_s) - \frac{1}{n}\boldsymbol{h}_v^1(\mathcal{G}') + (\frac{1}{n}-\frac{1}{n'})\boldsymbol{h}_v^1(\mathcal{G}'))\right|\right)$$

$$\leq L_1 LM \left(\left|\frac{1}{n}(\boldsymbol{h}_u^1(\mathcal{G}_s) - \boldsymbol{h}_u^1(\mathcal{G}'))\right| + \left|(\frac{1}{n}-\frac{1}{n'})\boldsymbol{h}_u^1(\mathcal{G}')\right| + \left|\sum_{v\in\mathcal{N}_u(\mathcal{G}_s)\backslash\mathcal{N}_u(\mathcal{G}')}\frac{1}{n}\sum_{p\in\mathcal{N}_v(\mathcal{G}_s)}\frac{1}{m}\boldsymbol{h}_p^0(\mathcal{G}_s)\right|\right.$$

$$+ \left|\sum_{v\in\mathcal{N}_u(\mathcal{G}')\backslash\mathcal{N}_u(\mathcal{G}_s)}\frac{1}{n'}\sum_{p\in\mathcal{N}_v(\mathcal{G}')}\frac{1}{m'}\boldsymbol{h}_p^0(\mathcal{G}')\right|$$

$$+ \left.\left|\sum_{v\in\mathcal{N}_u(\mathcal{G}')\cap\mathcal{N}_u(\mathcal{G}_s)}\frac{1}{n}(\boldsymbol{h}_v^1(\mathcal{G}_s) - \boldsymbol{h}_v^1(\mathcal{G}')) + (\frac{1}{n}-\frac{1}{n'})\boldsymbol{h}_v^1(\mathcal{G}'))\right|\right)$$

$$\leq L_1 L^2 M^2 C\left(4\beta_s + \frac{3(1-\alpha_s)}{\alpha_s} + \beta_s + 2\beta' + 2\beta' + \frac{3(1-\alpha_s)}{\alpha_s} + \beta_s\right)$$

$$\leq L_1 L^2 M^2 C\left(2\left(4\beta_s + \frac{3(1-\alpha_s)}{\alpha_s}\right) + 4\beta'\right)$$

Thus,

$$|\mathsf{Pr}_u(\mathcal{G}) - \mathsf{Pr}_u(\mathcal{G}')|$$
$$\leq |\mathsf{Pr}_u(\mathcal{G}) - \mathsf{Pr}_u(\mathcal{G}_s)| + |\mathsf{Pr}_u(\mathcal{G}_s) - \mathsf{Pr}_u(\mathcal{G}')|$$
$$\leq L_1 L^2 M^2 C\left(2\left(4\beta_s + \frac{3(1-\alpha_s)}{\alpha_s}\right) + 4\beta' + 2\right)$$

Similarly, If the GNN has the $T$ layer, then,

$$|\mathsf{Pr}_u(\mathcal{G}) - \mathsf{Pr}_u(\mathcal{G}')|$$
$$\leq \cdots$$
$$\leq L_1 L^2 M^2 C\left(2^{T-1}\left(4\beta_s + \frac{3(1-\alpha_s)}{\alpha_s}\right) + 4(T-1)\beta' + 2(T-1)\right).$$

$\square$

### C.4 Proof of Theorem 4.7

*Proof.* let $\mathcal{B}_t(u)(\mathcal{G}) := \{v \in \mathcal{V} : d_\mathcal{G}(u,v) \leq t\}$. $\bar{R}_u = \frac{\sum_{(q,v)\in\mathcal{E}} R_{q,v}}{|\mathcal{E}|}$,

**If $\bigoplus$ is the sum operation**, then,

$$|\mathrm{Pr}_u(\mathcal{G}) - \mathrm{Pr}_u(\mathcal{G}')| \leq L_1 \left| \boldsymbol{h}_u^T(\mathcal{G}) - \boldsymbol{h}_u^T(\mathcal{G}') \right|$$

$$\leq L_1 L M \left| \sum_{v \in \mathcal{N}_u(\mathcal{G})} A_{vu} \boldsymbol{h}_v^{T-1}(\mathcal{G}) - \sum_{v \in \mathcal{N}_u(\mathcal{G}')} A'_{vu} \boldsymbol{h}_v^{T-1}(\mathcal{G}') \right|$$

$$\leq \cdots \cdots$$

$$\leq L_1 L^T M^T \left| \sum_{p \in \mathcal{N}_j(\mathcal{G})} \cdots \sum_{v \in \mathcal{N}_u(\mathcal{G})} A_{pj} \cdots A_{vu} \boldsymbol{h}_p^0(\mathcal{G}) - \sum_{p \in \mathcal{N}_j(\mathcal{G}')} \cdots \sum_{v \in \mathcal{N}_u(\mathcal{G}')} A'_{pj} \cdots A'_{vu} \boldsymbol{h}_i^{T-2}(\mathcal{G}') \right|$$

$$\leq L_1 L^T M^T C \left| \sum_{p \in \mathcal{B}_T(u)(\mathcal{G})} A_{pu}^T - \sum_{p \in \mathcal{B}_T(u)(\mathcal{G}')} A_{pu}'^T \right|$$

$$\leq L_1 L^T M^T C \left| \sum_{p \in \mathcal{B}_T(u)(\mathcal{G})} A_{pu}^T - \sum_{p \in \mathcal{B}_T(u)(\mathcal{G}')} A_{pu}'^T + \bar{R}_u - \bar{R}_u \right|$$

$$\leq \left| \sum_{p \in \mathcal{B}_T(u)(\mathcal{G})} A_{pu}^T - \sum_{p \in \mathcal{B}_T(u)(\mathcal{G}')} A_{pu}'^T + \bar{R}_u \right|$$

$$+ \left| \frac{\sum_{(q,v) \in \mathcal{V}} \left( \frac{1}{\sqrt{d_q}} 1_q - \frac{1}{\sqrt{d_v}} 1_v \right)^\top \hat{L}^+ \left( \frac{1}{\sqrt{d_q}} 1_q - \frac{1}{\sqrt{d_v}} 1_v \right)}{|\mathcal{V}|} \right|$$

**If $\bigoplus$ is the mean operation**, then,

$$|\mathrm{Pr}_u(\mathcal{G}) - \mathrm{Pr}_u(\mathcal{G}')| \leq L_1 \left| \boldsymbol{h}_u^T(\mathcal{G}) - \boldsymbol{h}_u^T(\mathcal{G}') \right|$$

$$\leq L_1 L M \left| \sum_{v \in \mathcal{N}_u(\mathcal{G})} \hat{A}_{vu} \boldsymbol{h}_v^{T-1}(\mathcal{G}) - \sum_{v \in \mathcal{N}_u(\mathcal{G}')} \hat{A}'_{vu} \boldsymbol{h}_v^{T-1}(\mathcal{G}') \right|$$

$$\leq L_1 L^2 M^2 \left| \sum_{i \in \mathcal{N}_v(\mathcal{G})} \sum_{v \in \mathcal{N}_u(\mathcal{G})} \hat{A}_{iv} \hat{A}_{vu} \boldsymbol{h}_i^{T-2}(\mathcal{G}) - \sum_{i \in \mathcal{N}_v(\mathcal{G}')} \sum_{v \in \mathcal{N}_u(\mathcal{G}')} \hat{A}'_{iv} \hat{A}'_{vu} \boldsymbol{h}_i^{T-2}(\mathcal{G}') \right|$$

$$\leq \cdots \cdots$$

$$\leq L_1 L^T M^T \left| \sum_{p \in \mathcal{N}_j(\mathcal{G})} \cdots \sum_{v \in \mathcal{N}_u(\mathcal{G})} \hat{A}_{pj} \cdots \hat{A}_{vu} \boldsymbol{h}_p^0(\mathcal{G}) - \sum_{p \in \mathcal{N}_j(\mathcal{G}')} \cdots \sum_{v \in \mathcal{N}_u(\mathcal{G}')} \hat{A}'_{pj} \cdots \hat{A}'_{vu} \boldsymbol{h}_i^{T-2}(\mathcal{G}') \right|$$

$$\leq L_1 L^T M^T C \left| \sum_{p \in \mathcal{B}_T(u)(\mathcal{G})} \hat{A}_{pu}^T - \sum_{p \in \mathcal{B}_T(u)(\mathcal{G}')} \hat{A}_{pu}'^T \right|$$

$$\leq L_1 L^T M^T C \left| \sum_{p \in \mathcal{B}_T(u)(\mathcal{G})} \hat{A}_{pu}^T - \sum_{p \in \mathcal{B}_T(u)(\mathcal{G}')} \hat{A}_{pu}'^T + \bar{R}_u - \bar{R}_u \right|$$

$$\leq \left| \sum_{p \in \mathcal{B}_T(u)(\mathcal{G})} \hat{A}_{pu}^T - \sum_{p \in \mathcal{B}_T(u)(\mathcal{G}')} \hat{A}_{pu}'^T + \bar{R}_u \right|$$

$$+ \left| \frac{\sum_{(q,v) \in \mathcal{V}} \left( \frac{1}{\sqrt{d_q}} 1_q - \frac{1}{\sqrt{d_v}} 1_v \right)^\top \hat{L}^+ \left( \frac{1}{\sqrt{d_q}} 1_q - \frac{1}{\sqrt{d_v}} 1_v \right)}{|\mathcal{V}|} \right|$$

Let $\gamma$ denote the maximum eigenvalue of $\hat{A}$. We can bound the last term in the above equation using the *Courant-Fischer Theorem*, which says for a symmetric matrix $B$ with maximum eigenvalue $\gamma$ and any vector $x$, one has that $x^T B x \le x^T x \cdot |\gamma|$. Then, we have that

$$\left| \frac{\sum_{(q,v)\in\mathcal{V}} \left( \frac{1}{\sqrt{d_q}} 1_q - \frac{1}{\sqrt{d_v}} 1_v \right)^{\top} \hat{L}^+ \left( \frac{1}{\sqrt{d_q}} 1_q - \frac{1}{\sqrt{d_v}} 1_v \right)}{|\mathcal{V}|} \right|$$

$$\le \left| \frac{\sum_{(q,v)\in\mathcal{V}}(\frac{1}{d_q} + \frac{1}{d_v}) \sum_{t=1}^{\infty} \gamma^t}{|\mathcal{V}|} \right|$$

$$\le \left| \frac{\sum_{(q,v)\in\mathcal{V}}(\frac{1}{d_q} + \frac{1}{d_v})\frac{1}{1-\gamma}}{|\mathcal{V}|} \right| \qquad (\text{as } \gamma \in (-1,1))$$

$$\le \frac{2}{d_{min}} \frac{1}{1-\gamma}$$

Thus, whether $\bigoplus$ is the mean or sum operation, we can obtain the following inequality,

$$|\mathsf{Pr}_u(\mathcal{G}) - \mathsf{Pr}_u(\mathcal{G}')| \le L_1 L^T M^T C \left( 2|\mathcal{V}| + \bar{R}_u + \frac{2}{d_{min}} \frac{1}{1-\gamma} \right)$$

$\square$

## C.5  Proof of Theorem 4.8

*Proof.* Let $\mathcal{B}_t(u)(\mathcal{G}_s) := \{ v \in \mathcal{V}_s : d_{\mathcal{G}}(u,v) \le t \}$. $\bar{R}_s = \frac{\sum_{(q,v)\in\mathcal{E}_s} R_{q,v}}{|\mathcal{E}_s|}$.

$$|\mathsf{Pr}_u(\mathcal{G}) - \mathsf{Pr}_u(\mathcal{G}')| \le |\mathsf{Pr}_u(\mathcal{G}) - \mathsf{Pr}_u(\mathcal{G}_s)| + |\mathsf{Pr}_u(\mathcal{G}_s) - \mathsf{Pr}_u(\mathcal{G}')|$$

**If $\bigoplus$ is the sum operation**, then,

$$|\mathsf{Pr}_u(\mathcal{G}_s) - \mathsf{Pr}_u(\mathcal{G}')| \le L_1 \left| \boldsymbol{h}_u^T(\mathcal{G}_s) - \boldsymbol{h}_u^T(\mathcal{G}') \right|$$

$$\le L_1 L M \left| \sum_{v\in\mathcal{N}_u(\mathcal{G}_s)} A_{vu} \boldsymbol{h}_v^{T-1}(\mathcal{G}_s) - \sum_{v\in\mathcal{N}_u(\mathcal{G}')} A'_{vu} \boldsymbol{h}_v^{T-1}(\mathcal{G}') \right|$$

$$\le L_1 L^2 M^2 \left| \sum_{i\in\mathcal{N}_v(\mathcal{G}_s)} \sum_{v\in\mathcal{N}_u(\mathcal{G}_s)} A_{iv} A_{vu} \boldsymbol{h}_i^{T-2}(\mathcal{G}_s) - \sum_{i\in\mathcal{N}_v(\mathcal{G}')} \sum_{v\in\mathcal{N}_u(\mathcal{G}')} A'_{iv} A'_{vu} \boldsymbol{h}_i^{T-2}(\mathcal{G}') \right|$$

$$\le \cdots\cdots$$

$$\le L_1 L^T M^T \left| \sum_{p\in\mathcal{N}_j(\mathcal{G}_s)} \cdots \sum_{v\in\mathcal{N}_u(\mathcal{G}_s)} A_{pj} \cdots A_{vu} \boldsymbol{h}_p^0(\mathcal{G}_s) - \sum_{p\in\mathcal{N}_j(\mathcal{G}')} \cdots \sum_{v\in\mathcal{N}_u(\mathcal{G}')} A'_{pj} \cdots A'_{vu} \boldsymbol{h}_i^{T-2}(\mathcal{G}') \right|$$

$$\le L_1 L^T M^T C \left| \sum_{p\in\mathcal{B}_T(u)(\mathcal{G}_s)} A_{pu}^T - \sum_{p\in\mathcal{B}_T(u)(\mathcal{G}')} A'^T_{pu} \right|$$

$$\le L_1 L^T M^T C \left| \sum_{p\in\mathcal{B}_T(u)(\mathcal{G}_s)} A_{pu}^T - \sum_{p\in\mathcal{B}_T(u)(\mathcal{G}')} A'^T_{pu} + \bar{R}_s - \bar{R}_s \right|$$

$$\le \left| \sum_{p\in\mathcal{B}_T(u)(\mathcal{G}_s)} A_{pu}^T - \sum_{p\in\mathcal{B}_T(u)(\mathcal{G}')} A'^T_{pu} + \bar{R}_s \right|$$

$$+ \left| \frac{\sum_{(q,v)\in\mathcal{V}} \left( \frac{1}{\sqrt{d_q}} 1_q - \frac{1}{\sqrt{d_v}} 1_v \right)^{\top} \hat{L}^+ \left( \frac{1}{\sqrt{d_q}} 1_q - \frac{1}{\sqrt{d_v}} 1_v \right)}{|\mathcal{V}|} \right|$$

**If $\bigoplus$ is the mean operation**, then,

$$|\mathsf{Pr}_u(\mathcal{G}_s) - \mathsf{Pr}_u(\mathcal{G}')| \le L_1 \left| \boldsymbol{h}_u^T(\mathcal{G}_s) - \boldsymbol{h}_u^T(\mathcal{G}') \right|$$

$$\le L_1 L M \left| \sum_{v \in \mathcal{N}_u(\mathcal{G}_s)} \hat{A}_{vu} \boldsymbol{h}_v^{T-1}(\mathcal{G}_s) - \sum_{v \in \mathcal{N}_u(\mathcal{G}')} \hat{A}'_{vu} \boldsymbol{h}_v^{T-1}(\mathcal{G}') \right|$$

$$\le L_1 L^2 M^2 \left| \sum_{i \in \mathcal{N}_v(\mathcal{G}_s)} \sum_{v \in \mathcal{N}_u(\mathcal{G}_s)} \hat{A}_{iv} \hat{A}_{vu} \boldsymbol{h}_i^{T-2}(\mathcal{G}_s) - \sum_{i \in \mathcal{N}_v(\mathcal{G}')} \sum_{v \in \mathcal{N}_u(\mathcal{G}')} \hat{A}'_{iv} \hat{A}'_{vu} \boldsymbol{h}_i^{T-2}(\mathcal{G}') \right|$$

$$\le \cdots\cdots$$

$$\le L_1 L^T M^T \left| \sum_{p \in \mathcal{N}_j(\mathcal{G}_s)} \cdots \sum_{v \in \mathcal{N}_u(\mathcal{G}_s)} \hat{A}_{pj} \cdots \hat{A}_{iv} \hat{A}_{vu} \boldsymbol{h}_p^0(\mathcal{G}_s) - \sum_{p \in \mathcal{N}_j(\mathcal{G}')} \cdots \sum_{v \in \mathcal{N}_u(\mathcal{G}')} \hat{A}'_{pj} \cdots \hat{A}'_{iv} \hat{A}'_{vu} \boldsymbol{h}_i^{T-2}(\mathcal{G}') \right|$$

$$\le L_1 L^T M^T C \left| \sum_{p \in \mathcal{B}_T(u)(\mathcal{G}_s)} \hat{A}_{pu}^T - \sum_{p \in \mathcal{B}_T(u)(\mathcal{G}')} \hat{A}_{pu}'^T \right|$$

$$\le L_1 L^T M^T C \left| \sum_{p \in \mathcal{B}_T(u)(\mathcal{G}_s)} \hat{A}_{pu}^T - \sum_{p \in \mathcal{B}_T(u)(\mathcal{G}')} \hat{A}_{pu}'^T + \bar{R}_s - \bar{R}_s \right|$$

$$\le \left| \sum_{p \in \mathcal{B}_T(u)(\mathcal{G}_s)} \hat{A}_{pu}^T - \sum_{p \in \mathcal{B}_T(u)(\mathcal{G}')} \hat{A}_{pu}'^T + \bar{R}_s \right|$$

$$+ \left| \frac{\sum_{(q,v)\in\mathcal{V}_s} \left( \frac{1}{\sqrt{d_q}} 1_q - \frac{1}{\sqrt{d_v}} 1_v \right)^\top \hat{L}^+ \left( \frac{1}{\sqrt{d_q}} 1_q - \frac{1}{\sqrt{d_v}} 1_v \right)}{|\mathcal{V}|} \right|$$

Let $\gamma$ denote the maximum eigenvalue of $\hat{A}$. We can bound the last term in the above equation using the *Courant-Fischer Theorem*, which says for a symmetric matrix $B$ with maximum eigenvalue $\gamma$ and any vector $x$, one has that $x^T B x \le x^T x \cdot |\gamma|$. Then, we have that,

$$\left| \frac{\sum_{(q,v)\in\mathcal{V}} \left( \frac{1}{\sqrt{d_q}} 1_q - \frac{1}{\sqrt{d_v}} 1_v \right)^\top \hat{L}^+ \left( \frac{1}{\sqrt{d_q}} 1_q - \frac{1}{\sqrt{d_v}} 1_v \right)}{|\mathcal{V}|} \right|$$

$$\le \left| \frac{\sum_{(q,v)\in\mathcal{V}} (\frac{1}{d_q} + \frac{1}{d_v}) \sum_{t=1}^\infty \gamma^t}{|\mathcal{V}|} \right|$$

$$\le \left| \frac{\sum_{(q,v)\in\mathcal{V}} (\frac{1}{d_q} + \frac{1}{d_v}) \frac{1}{1-\gamma}}{|\mathcal{V}|} \right| \qquad \text{(as } \gamma \in (-1,1))$$

$$\le \frac{2}{d_{min}} \frac{1}{1-\gamma}$$

Then,

$$|\mathsf{Pr}_u(\mathcal{G}_s) - \mathsf{Pr}_u(\mathcal{G}')| \le L_1 L^T M^T C \left( 2|\mathcal{V}| + \bar{R}_s + \frac{2}{d_{min}} \frac{1}{1-\gamma} \right)$$

Thus,

$$|\mathsf{Pr}_u(\mathcal{G}) - \mathsf{Pr}_u(\mathcal{G}')| \le L_1 L^T M^T C \left( 2|\mathcal{V}| + \bar{R}_s + \frac{2}{d_{min}} \frac{1}{1-\gamma} + 2 \right)$$

$\square$

# D  Experiments

## D.1  Datasets

**Node classification datasets**: Cora, Citeseer, and PubMed [1] are citation networks. Each node represents a paper with a bag-of-words feature vector, connected by citation relationships. The task is to predict each paper's research area. **Link prediction datasets**: BC-OTC and BC-Alpha are trust networks of Bitcoin users on a trading platform. UCI is a social network of University of California, Irvine students, where links represent sent messages between users. **Graph classification datasets**: MUTAG [44] represents atom graph, with edges between bounding atoms. PROTEINS represents protein structures, where nodes are amino acids, and edges form if the distance between nodes is less than 6 Å apart. The graph label indicates whether the protein is an enzyme. IMDB-BINARY is movie collaboration dataset. Each graph corresponds to an ego-network for each actor/actress, where nodes correspond to actors/actresses and an edges indicate two actors/actresses co-appearances in movies. The details of data are in Table 9.

Table 9: The details of datasets

| Datasets | Nodes(Avg. Nodes) | Edges(Avg. Edges) | task |
|---|---|---|---|
| Cora | 2,708 | 10,556 | node classification |
| Citeseer | 3,321 | 9,196 | node classification |
| PubMed | 19,717 | 44,324 | node classification |
| BC-OTC | 5,881 | 35,588 | link prediction |
| BC-Alpha | 3,777 | 24,173 | link prediction |
| UCI | 1,899 | 59,835 | link prediction |
| MUTAG | 17.93 | 39.6 | graph classification |
| PROTEINS | 39.1 | 145.6 | graph classification |
| IMDB-BINARY | 19.8 | 193.1 | graph classification |

## D.2  GNN models

We trained two 2-layer GNNs, one with element-wise sum and the other with mean as the aggregation function. For node classification, $h_u(G)$ is mapped to the class distribution through the softmax function. For the link prediction, we concatenate $h_u(G)$ and $h_v(G)$ as the input to a linear layer to obtain the logits, which are then mapped to the probability of the existence of the edge $(u, v)$. For the graph classification task, the average pooling of $h_u(G)$ across all nodes in $G$ can produce a single vector representation $h(G)$ for classification. It is mapped to the class probability distribution through the softmax function. During training, we set the learning rate to 0.01, the dropout rate to 0.5 and the hidden size to 16. The model is trained and then fixed during the explanation stages. Our experiments were done on a CPU with a kernel size of 32GB.

## D.3  Quantitative evaluation metrics

For the night datasets, we randomly select nodes, edges, and graphs as target sets. For each target, given a sparsity level of 0.9, we apply our method to generate robust explanations. As baselines, we include random explanations (via randomly selected edges) and base explanations produced by six existing GNN explanation methods. We then create 300 perturbed graphs to evaluate $\delta_*$ for four types of explanations: base, random, Ricci curvature-based, and effective resistance-based. For each target, we compute the relative error and Fidelity$_{KL}$ and report their averages over all target sets.

## D.4  Explanation Methods

**GNNExplainer** learns edges masks by maximizing the mutual information to explain GNN predictions. **PGExplainer** learns approximated discrete masks for edges to explain the predictions. **GNN-LRP** utilizes the LRP back-propagation attribution method to GNN, attributing class probability to input neurons. **DeepLIFT** attributes the log-odd between two probabilities and uses a summation function to obtain contributions of edges. **FlowX** applies the Shapley value to derive initial contributions of message flows, then trains these scores with loss functions and maps them to edges. **Convex** designs a convex objective function to approximate the KL divergence and obtains the important edges by solving convex optimization.

## D.5 The details of three large datasets

For the Pubmed, Coauthor-Computer, and Coauthor-Physics datasets, the statistics are summarized in Table 10.

Table 10: Three large graph datasets

| Datasets | Classes | Nodes | Edges | Edge/Node | Features |
|---|---|---|---|---|---|
| PubMed | 3 | 19,717 | 44,324 | 2.24 | 500 |
| Coauthor-Computer | 13 | 18,333 | 327,576 | 17.87 | 6,805 |
| Coauthor-Physics | 2 | 34,493 | 991,848 | 28.76 | 8,415 |

## D.6 The $\lambda$ values across methods and datasets

When the $\bigoplus$ is the mean function, the selected $\lambda$ values for Ricci curvature and effective resistance across methods and datasets are reported in Tables 11 and 12, respectively. When $\bigoplus$ is the sum function, the corresponding values are shown in Tables 13 and 14.

Table 11: Selected $\lambda$ values for robust explanations based on **Ricci curvature**. The $\bigoplus$ in GNN is **mean** operation.

| Methods | Cora | Citeseer | Pubmed | BC-OTC | BC-Alpha | UCI | MUTAG | PROTEINS | IMDB-BINARY |
|---|---|---|---|---|---|---|---|---|---|
| GNNExplainer | 0.782 | 0.487 | 0.035 | 0.024 | 0.067 | 0.009 | 0.004 | 0.223 | 0.784 |
| PGExplainer | 0.565 | 0.671 | 0.038 | 0.879 | 0.344 | 0.789 | 0.288 | 0.003 | 0.006 |
| Convex | 0.748 | 0.982 | 0.652 | 0.095 | 0.510 | 0.798 | 0.499 | 0.813 | 0.397 |
| DeepLIFT | 0.116 | 0.016 | 0.077 | 0.792 | 0.535 | 0.519 | 0.029 | 0.689 | 0.149 |
| GNN-LRP | 0.291 | 0.182 | 0.261 | 0.001 | 0.143 | 0.816 | 0.481 | 0.749 | 0.225 |
| FlowX | 0.030 | 0.001 | 0.002 | 0.176 | 0.323 | 0.704 | 0.077 | 0.491 | 0.062 |

Table 12: Selected $\lambda$ values for robust explanations based on **effective resistance**. The $\bigoplus$ in GNN is **mean** operation.

| Methods | Cora | Citeseer | Pubmed | BC-OTC | BC-Alpha | UCI | MUTAG | PROTEINS | IMDB-BINARY |
|---|---|---|---|---|---|---|---|---|---|
| GNNExplainer | 0.949 | 0.964 | 0.925 | 0.05 | 0.04 | 0.035 | 0.6 | 0.889 | 0.69 |
| PGExplainer | 0.101 | 0.074 | 0.059 | 0.424 | 0.8 | 0.523 | 0.983 | 0.483 | 0.529 |
| Convex | 0.581 | 0.226 | 0.524 | 0.052 | 0.129 | 0.984 | 0.082 | 0.226 | 0.547 |
| DeepLIFT | 0.011 | 0.010 | 0.006 | 0.096 | 0.002 | 0.021 | 0.462 | 0.739 | 0.148 |
| GNN-LRP | 0.056 | 0.052 | 0.037 | 0.001 | 0.162 | 0.111 | 0.432 | 0.490 | 0.620 |
| FlowX | 0.091 | 0.072 | 0.013 | 0.011 | 0.062 | 0.112 | 0.387 | 0.040 | 0.200 |

## D.7 Running time

We report the running time for computing Ricci curvature and effective resistance on the Pubmed, Coauthor-Computer, and Coauthor-Physics datasets (see Table 10). Results are shown in Figures 3 and 4. As illustrated in Figure 3. For 15,000 edges, the computation of Ricci curvature is very time-efficient, taking only 3.5 seconds. In contrast, computing effective resistance is more time-consuming, taking approximately 60 seconds for the same number of edges. Nevertheless, the overall computation time remains acceptable for practical use.

## D.8 Ablation analysis

According to the Figures 5, 6, 7, and 8, we can see that even if $\lambda$ changes, on these datasets, using Ricci curvature and effective resistance can achieve better robustness and fidelity.

Table 13: Selected $\lambda$ values for robust explanations based on **Ricci curvature**. The $\bigoplus$ in GNN is **sum** operation.

| Methods | Cora | Citeseer | Pubmed | BC-OTC | BC-Alpha | UCI | MUTAG | PROTEINS | IMDB-BINARY |
|---|---|---|---|---|---|---|---|---|---|
| **GNNExplainer** | 0.017 | 0.035 | 0.048 | 0.080 | 0.017 | 0.022 | 0.654 | 0.001 | 0.147 |
| **PGExplainer** | 0.871 | 0.188 | 0.194 | 0.687 | 0.583 | 0.538 | 0.654 | 0.92 | 0.292 |
| **Convex** | 0.022 | 0.04 | 0.084 | 0.822 | 0.295 | 0.470 | 0.513 | 0.813 | 0.792 |
| **DeepLIFT** | 0.001 | 0.009 | 0.03 | 0.317 | 0.154 | 0.001 | 0.589 | 0.046 | 0.379 |
| **GNNLRP** | 0.049 | 0.042 | 0.032 | 0.001 | 0.357 | 0.001 | 0.708 | 0.022 | 0.086 |
| **FlowX** | 0.01 | 0.05 | 0.017 | 0.001 | 0.001 | 0.007 | 0.023 | 0.939 | 0.660 |

Table 14: Selected $\lambda$ values for robust explanations based on **effective resistance**. The $\bigoplus$ in GNN is **sum** operation.

| Methods | Cora | Citeseer | Pubmed | BC-OTC | BC-Alpha | UCI | MUTAG | PROTEINS | IMDB-BINARY |
|---|---|---|---|---|---|---|---|---|---|
| **GNNExplainer** | 0.200 | 0.008 | 0.677 | 0.033 | 0.011 | 0.066 | 0.294 | 0.007 | 0.644 |
| **PGExplainer** | 0.069 | 0.005 | 0.149 | 0.001 | 0.003 | 0.001 | 0.982 | 0.262 | 0.005 |
| **Convex** | 0.65 | 0.35 | 0.637 | 0.795 | 0.001 | 0.942 | 0.101 | 0.110 | 0.255 |
| **DeepLIFT** | 0.002 | 0.002 | 0.025 | 0.072 | 0.029 | 0.028 | 0.785 | 0.032 | 0.650 |
| **GNN-LRP** | 0.009 | 0.142 | 0.037 | 0.012 | 0.086 | 0.442 | 0.859 | 0.050 | 0.990 |
| **FlowX** | 0.25 | 0.005 | 0.005 | 0.095 | 0.070 | 0.472 | 0.629 | 0.015 | 0.291 |

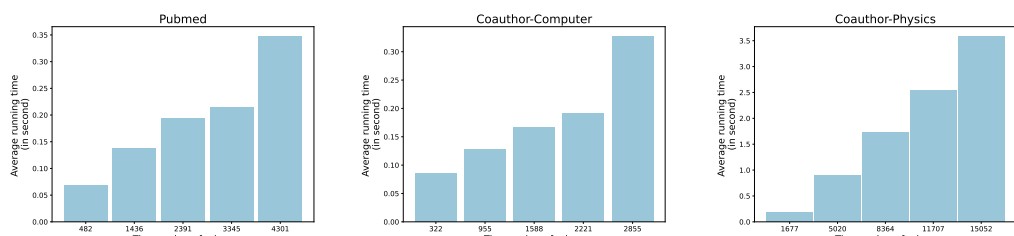

Figure 3: The running time of calculating the Ricci curvature of edges.

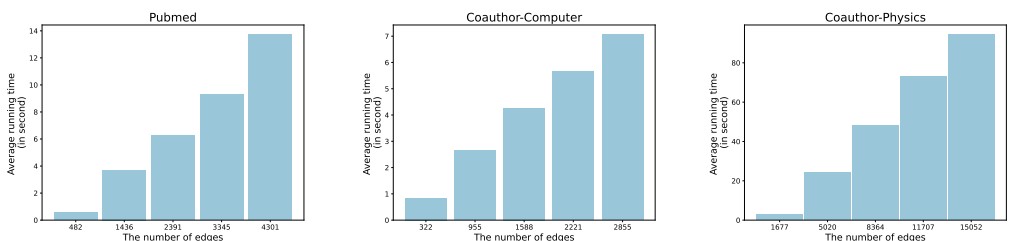

Figure 4: The running time of calculating the effective resistance of edges.

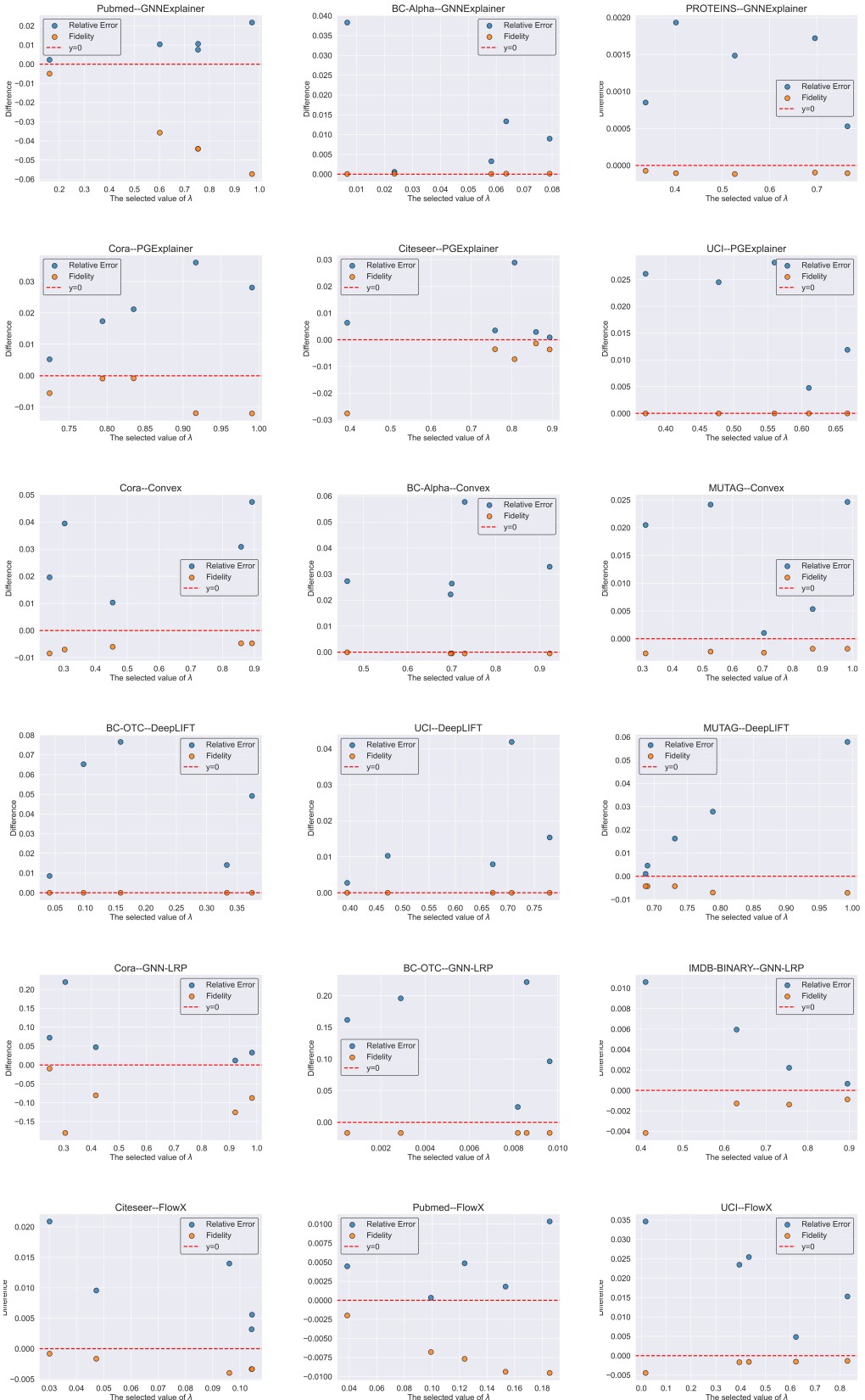

Figure 5: The ⊕ in GNN is **mean** operation. Rows represent explanation methods, columns denote datasets, and the x-axis indicates the selected value of $\lambda$. The y-axis denotes the difference in relative error or fidelity between the **Ricci curvature** based and original explanations.

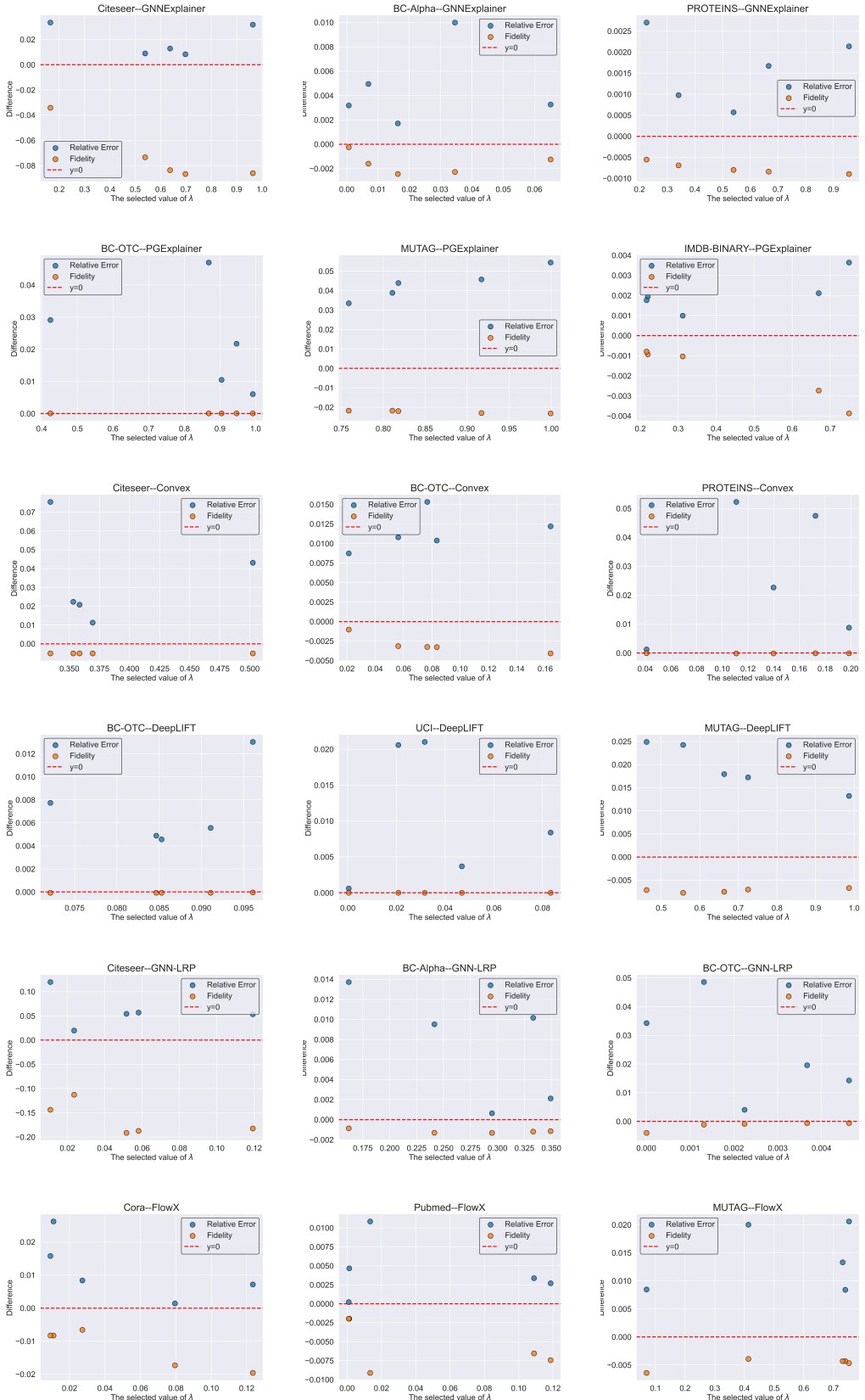

Figure 6: The ⊕ in GNN is **mean** operation. Rows represent explanation methods, columns denote datasets, and the x-axis indicates the selected value of $\lambda$. The y-axis denotes the difference in relative error or fidelity between the **effective resistance** based and original explanations.

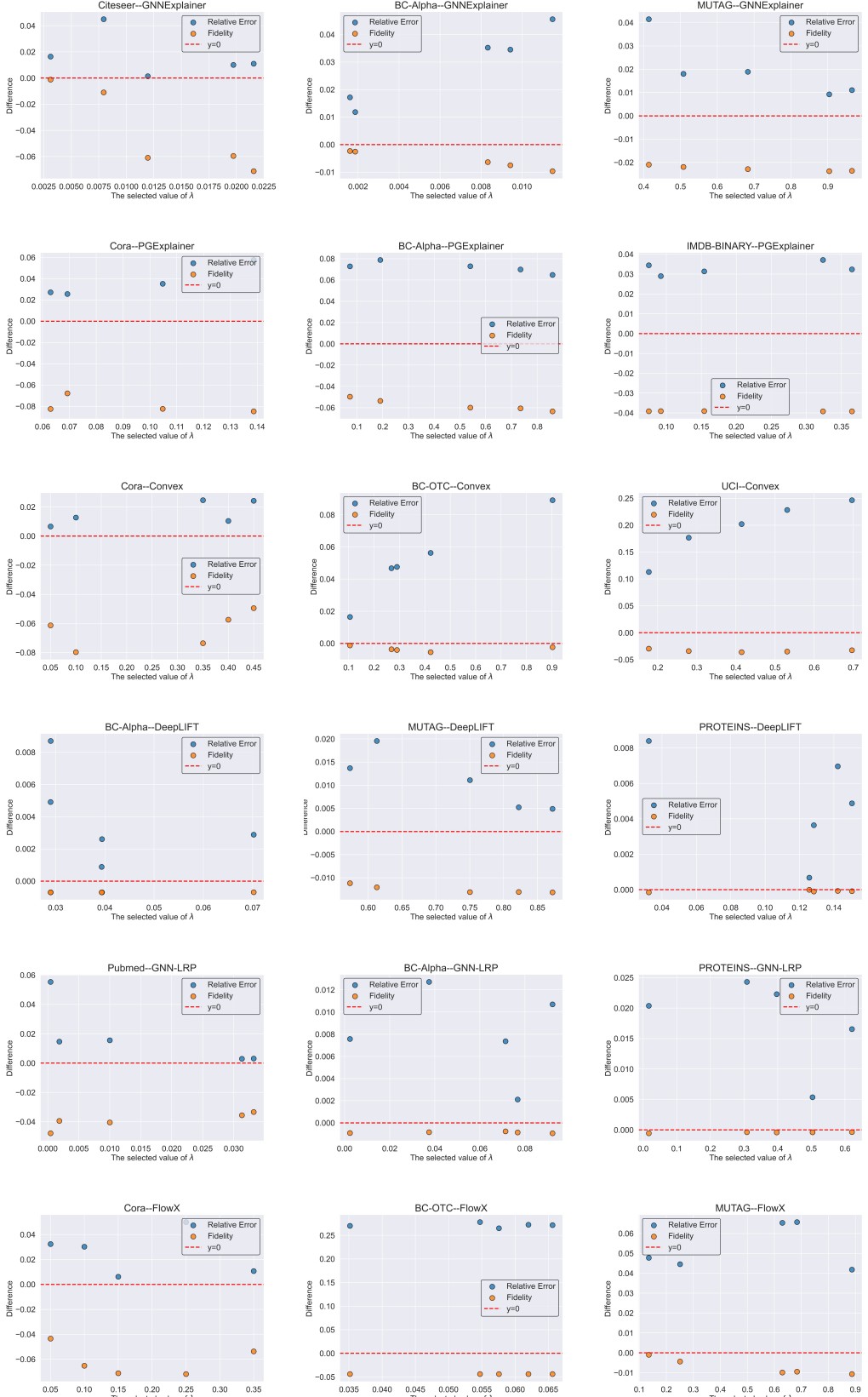

Figure 7: The $\bigoplus$ in GNN is **sum** operation. Rows represent explanation methods, columns denote datasets, and the x-axis indicates the selected value of $\lambda$. The y-axis denotes the difference in relative error or fidelity between the **effective resistance** based and original explanations.

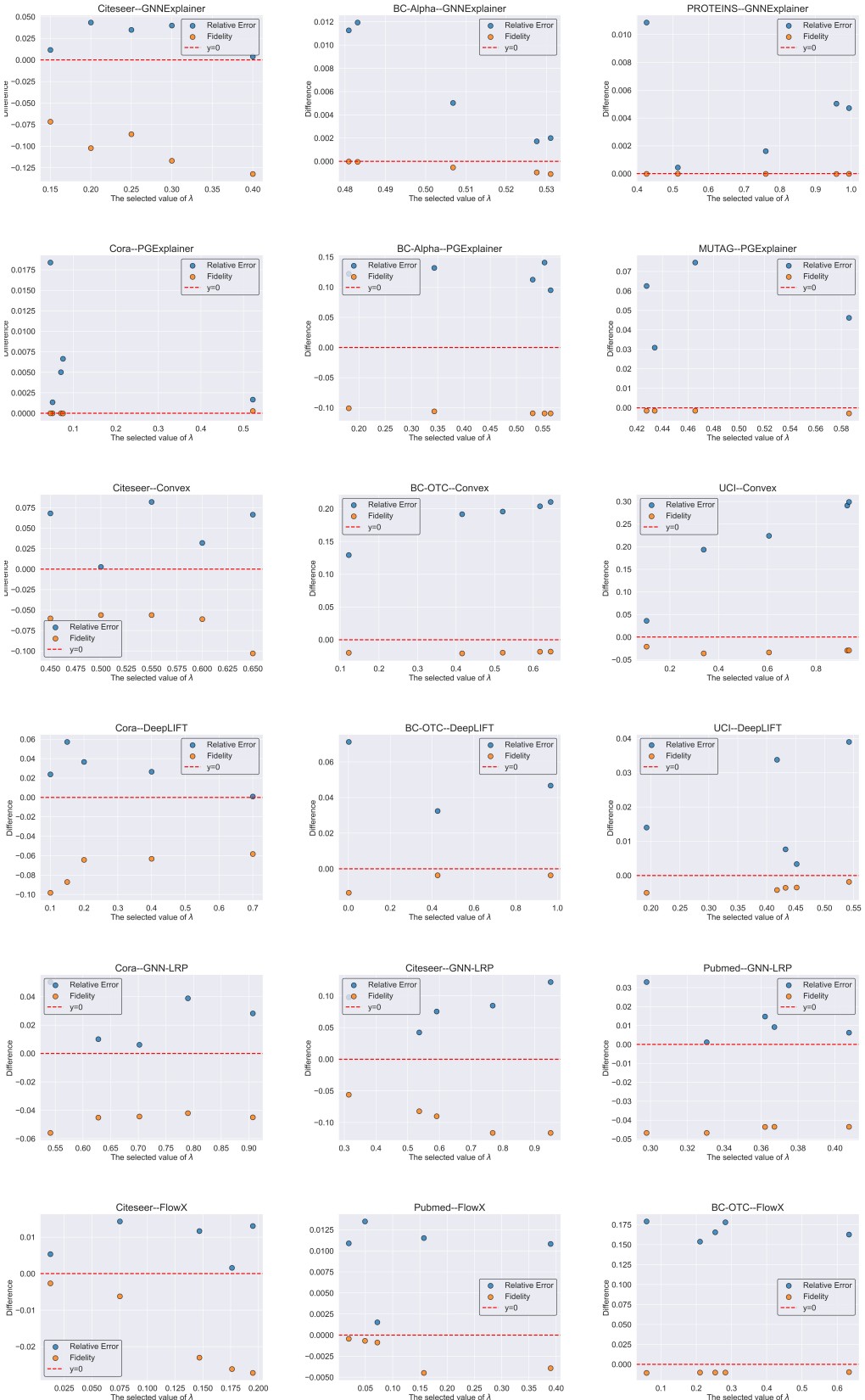

Figure 8: The ⊕ in GNN is **sum** operation. Rows represent explanation methods, columns denote datasets, and the x-axis indicates the selected value of $\lambda$. The y-axis denotes the difference in relative error or fidelity between the **Ricci curvature** based and original explanations.

