# OpenReview forum: "Robust Explanations of Graph Neural Networks via Graph Curvatures"
_NeurIPS.cc/2025/Conference — NeurIPS 2025 poster_

### Official Review · Reviewer_Eh2L · 2025-06-05

**Clarity:** 3
**Significance:** 2
**Originality:** 3
**Rating:** 5
**Confidence:** 2

**Summary:**

This paper adds an auxiliary loss on the classic GNN explanation objective, enhancing the robustness of basic explanations.
Specifically, the authors adopt the two concepts of Ricci curvature and effective resistance in graph theory to derive the auxiliary loss respectively, and theoretically draw a connection to explanation robustness.
Experiments show that the proposed loss function consistently improves the robustness of GNN explanations while sacrificing less in terms of fidelity.

**Questions:**

Please refer to the weaknesses and limitations for suggestions.

**Ethical Concerns:**

["NO or VERY MINOR ethics concerns only"]

**Final Justification:**

Although the authors' empirical results are not strong enough, my main concern has been addressed. Considering that the authors' responses to other reviewers are comprehensive with additional results, I believe this paper meets the acceptance criteria.

**Limitations:**

Though the theory is well-established, the application value of this work may be limited: If our ultimate goal is robustness, why don't we transform the robustness objective function into a tractable loss ?  For example, using reparameterization or directly measuring the distance of the multivariate distribution for Eq. (2).

**Quality:**

4

**Strengths And Weaknesses:**

**Strengths:**
- The paper provides a fresh perspective on robust GNN explanation, with sufficient theoretical analysis.
- The method shows significant robustness gains across multiple tasks, improving various base explainers.
- The paper is well-organized. The authors have done a lot of work.

**Weaknesses:**
- The experimental results lack significance analysis: How many times was the algorithm run? Is the improvement significant?

---

> ### Author Rebuttal · Authors · 2025-07-30
>
> Thank you for taking the time to review our paper!
> >**Q1: How many times was the algorithm run?**
>
> **A1**: We run the base explanation algorithm only once to get the edge importance scores. Then, we generate the final explanation subgraph using the importance scores, Ricci curvature, and effective resistance. For robustness evaluation, we sample 300 random perturbation subgraphs and compute the robustness metric using Equation (2). For other metrics like fidelity, the algorithm also runs only once.
>
> >**Q2: Is the improvement significant?**
>
> **A2**: We did not perform formal statistical significance testing in this version of the paper. In our experiments, while some improvements over baselines are consistent, they are not always statistically significant across all datasets. We will consider adding significance analysis in future work.
>
> >**Q3: why don't we transform the robustness objective function into a tractable loss ? For example, using reparameterization or directly measuring the distance of the multivariate distribution for Eq. (2).**
>
> **A3**: We agree that transforming Eq. (2) into an optimizable loss is an interesting idea. However, it is not straightforward to implement for several reasons. First, it is difficult to define how to generate and update perturbation subgraphs based on the explanation subgraph during training. Second, most existing explanation methods rely on continuous edge/node masks, while the final explanation is discrete. This mismatch can lead to information loss and suboptimal performance. Instead, our method directly selects robust and important edges based on geometric properties of the graph, improving the robustness of the explanation without requiring a new loss function.

---

> > ### Comment · Reviewer_Eh2L · 2025-08-01
> >
> > Since this paper studies robust explanations, you need to ensure the reproducibility --- different initializations or random seeds will not affect your robustness conclusions. We suggest that you supplement statistical analysis in your next revision. This would bolster confidence in your experimental results. :)

---

> > > ### Author Response · Authors · 2025-08-02
> > > **The response to reviewer Eh2L**
> > >
> > > We sincerely thank the reviewer for raising this important point about reproducibility and statistical reliability.
> > >
> > >  Following your suggestion, we have conducted statistical analysis to evaluate the robustness of our results. Specifically, we performed t-tests between the robustness scores of the original explanation methods and our Ricci curvature-based and effective resistance-based methods. The detailed test results will be included in the appendix of the revised version. In summary: In Table 1 (the current version) , 28% of the comparisons show statistically significant differences (p < 0.05); In Table 3 (the current version), 24% of the comparisons are statistically significant (p < 0.05). Although some datasets do not show significant differences, this does not necessarily mean that our conclusions depend heavily on the choice of random seed or initialization.
> > >
> > > In fact, the random seed only influences the evaluation procedure, not the explanation itself.
> > > We do evaluate the robustness of explanations under graph perturbations. For a given explanation subgraph, we randomly add and remove 10% of the edges that are not part of the explanation to create a perturbed graph G′. We repeat this process 300 times and compute the robustness score as defined in Definition 4.1.
> > >
> > > We test the robustness of both our method (based on Ricci curvature) and the original explanation method on several datasets, using different random seeds.
> > > We measure the difference by subtracting the robustness score of our method from that of the baseline. Since lower robustness scores indicate better performance, a **positive** value means our method is more robust. The results are shown in the table below.
> > >
> > > **Table 1: The difference in robustness metric between our method and the baseline method under different random seeds.**
> > > | method      |          |      cora       |             |     |     BC-Alpha         |             |        |     MUTAG         |             |
> > > |-------------|--------------|-------------|-------------|--------------|--------------|--------------|--------------|--------------|--------------|
> > > |             | seed_1       | seed_2      | seed_3      | seed_1       | seed_2       | seed_3       | seed_1       | seed_2       | seed_3       |
> > > | gnexplainer | 4.3%         | 4.5%        | 3.8%        | 1.3%         | 2.3%         | 1.1%         | 1.3%         | 0.6%         | 1.2%         |
> > > | pgexplainer | 2.0%         | 1.8%        | 1.4%        | 2.2%         | 2.0%         | 2.1%         | 1.2%         | 0.8%         | 1.3%         |
> > > | convex      | 23.3%        | 21.3%       | 23.0%       | 5.7%         | 6.0%         | 5.7%         | 2.4%         | 2.7%         | 1.6%         |
> > > | deeplift    | 2.0%         | 5.7%        | 3.8%        | 2.8%         | 1.2%         | 1.7%         | 2.2%         | 2.1%         | 2.1%         |
> > > | gnnlrp      | 4.1%         | 5.4%        | 4.0%        | 3.7%         | 4.0%         | 4.4%         | 2.2%         | 1.1%         | 2.4%         |
> > > | flowx       | 2.5%         | 2.4%        | 1.9%        | 0.7%         | 1.1%         | 0.9%         | 0.4%         | 0.8%         | 0.6%         |
> > >
> > > Our method consistently outperforms the baseline, regardless of the random seed.
> > > For a fixed dataset and method, the difference varies little across different random seeds. This indicates that the random seed has minimal impact on our final conclusion.

---

> > > > ### Comment · Reviewer_Eh2L · 2025-08-03
> > > >
> > > > Judging from the results in your report, the empirical evidence is not strong enough, as less than 30% of the comparisons show significant differences. In addition, the random seed does not necessarily only affect the evaluation procedure; it can also influence the GNN being explained — the parameters of GNNs trained on the same data can vary. In real-world scenarios, we need to analyze GNNs across different tasks, and at this point, it is necessary to ensure that the explanation methods can work stably.
> > > >
> > > > Considering that the approach, which combines the two concepts of Ricci curvature and effective resistance in graph theory, is inspiring, the limitations I pointed out above will not affect my personal evaluation. I will finally take other reviewers' responses into account to make my Final Justification.

---

> > > > > ### Author Response · Authors · 2025-08-04
> > > > >
> > > > > Thank you for taking the time to review our rebuttal. We deeply thank your reading and understanding of our rebuttal.

---

### Official Review · Reviewer_MPaM · 2025-06-25

**Clarity:** 4
**Significance:** 3
**Originality:** 4
**Rating:** 4
**Confidence:** 3

**Summary:**

This paper addresses the enhancement of robustness in explainability for Graph Neural Networks (GNNs). Prior approaches to improving robustness, such as model retraining and ensemble-based explanation, have significant limitations: retraining alters the model itself. At the same time, ensemble methods often yield overly divergent explanations, leading to high uncertainty. In contrast, this paper proposes a novel explanation method that emphasizes robust structural properties by leveraging the graph's geometric characteristics. Specifically, it quantifies the robustness of node connectivity using Ollivier-Ricci curvature and effective resistance, and theoretically proves that these quantities upper-bound the robustness of model predictions and explanations. The authors incorporate these measures as robustness scores into the objective functions of various existing explanation methods. Through experiments on nine benchmark datasets, the results demonstrate that incorporating the proposed method improves fidelity in over 80% of cases and achieves up to a 10% gain in explanation robustness.

**Questions:**

1. Could you discuss the possibility of determining the trade-off parameter lambda between robustness and fidelity not globally for the entire graph, but locally for each explanatory target?

2. Could you elaborate on the potential situations where the approximation used in the Ricci curvature calculation, based solely on node degrees, may violate the underlying assumptions of the approximation?

**Ethical Concerns:**

["NO or VERY MINOR ethics concerns only"]

**Final Justification:**

My concern regarding the potential oversimplification of using a single global lambda for the entire graph, and the possibility of determining it locally for each explanatory target, is largely consistent with the authors’ perspective. The additional experiments addressing whether the proposed approximation holds in high-dimensional and non-Euclidean graph spaces have improved the reliability of the approach. Based on these considerations, I remain inclined toward recommending acceptance; however, I will keep my rating unchanged.

**Limitations:**

The paper does not appear to consider the possibility that the trade-off between robustness and fidelity may vary locally across different explanatory targets. Additionally, there seems to be no discussion on scenarios where the assumptions underlying the approximation of Ricci curvature, based solely on node degrees, may not hold.

**Paper Formatting Concerns:**

There are no particular paper formatting concerns.

**Quality:**

3

**Strengths And Weaknesses:**

Strengths:
The paper provides a theoretical proof that Ollivier-Ricci curvature and effective resistance can upper-bound the robustness of both model predictions and explanations. Building upon this, the authors incorporate these measures as robustness scores into the objective functions of various explanation methods, enabling explanations with guaranteed robustness for GNNs. This constitutes a significant contribution to the field of GNN explainability. Furthermore, the paper empirically validates the effectiveness of the proposed method across nine diverse datasets and multiple explanation techniques.

Weaknesses:
While the trade-off between robustness and fidelity is managed via a tunable parameter lambda, an elegant and theoretically sound idea, the method does not account for the inherent diversity of graph data. For example, even within node classification tasks, there may be cases where robust edges serve as the primary rationale for predictions, and others where fragile edges are more predictive. Therefore, using a single global lambda to represent this trade-off across an entire graph may be overly simplistic. A more nuanced discussion on this point could greatly enhance the persuasiveness and practical applicability of the proposed method.

Additionally, the Ricci curvature approximation relies solely on node degrees, which could be problematic. For instance, it is entirely plausible that two high-degree nodes are connected via a single bridge-like edge, and in such cases, the approximation may fail. This simplification appears to generalize the known result that, in one-dimensional real-valued distributions, the Wasserstein distance can be represented as the area between CDFs. However, whether this approximation holds in high-dimensional and non-Euclidean graph spaces warrants more careful examination.

---

> ### Author Rebuttal · Authors · 2025-07-30
>
> Thank you for taking the time to review our paper!
> >**Q1:While the trade-off between robustness and fidelity is managed via a tunable parameter lambda, an elegant and theoretically sound idea, the method does not account for the inherent diversity of graph data. For example, even within node classification tasks, there may be cases where robust edges serve as the primary rationale for predictions, and others where fragile edges are more predictive. Therefore, using a single global lambda to represent this trade-off across an entire graph may be overly simplistic.**
>
> **A1:** Thank you very much for raising this important question. We agree that the trade-off between fidelity and robustness can vary depending on the explanatory target. As the reviewer pointed out, in some cases, fragile edges may be essential for the model’s prediction, while robust edges may contribute less. In such cases, increasing lambda to favor robustness can reduce fidelity. However, even without using lambda, it may still be difficult to find a subgraph that is both robust and faithful using other methods. This may be due to the structure of the graph itself.
>
> On the other hand, in cases where many candidate edges have similar importance scores, using curvature-based robustness can help choose more stable edges and improve robustness without lowering fidelity.
>
> Our current experiments focus on global performance across all the explanation targets. Our results show that, in most cases, robustness can be improved without a significant drop in fidelity. We acknowledge that a more detailed analysis of target-specific trade-offs would be valuable.
>
> >**Q2: Could you discuss the possibility of determining the trade-off parameter lambda between robustness and fidelity not globally for the entire graph, but locally for each explanatory target?**
>
> **A2:** If it is a single target node, I think a more effective method may be to use grid search to obtain lambda. Using the optuan package in Python can speed up the search process.
>
> >**Q3: Additionally, the Ricci curvature approximation relies solely on node degrees, which could be problematic. For instance, it is entirely plausible that two high-degree nodes are connected via a single bridge-like edge, and in such cases, the approximation may fail. However, whether this approximation holds in high-dimensional and non-Euclidean graph spaces warrants more careful examination.**
>
> **A3:** We thank the reviewer for this insightful comment. Indeed, as pointed out, our original approximation relies solely on node degrees, neglecting explicit distance information within the graph. This simplification can indeed fail in specific scenarios, such as the one raised by the reviewer, where two high-degree nodes are connected via a single bridge-like edge. However, our experimental results demonstrate the effectiveness of this approximation method.
>
> To address this limitation and strengthen our results, we have further derived the theory using the exact Wasserstein distance, explicitly incorporating graph distances.  The theorem 4.4 become as
>
> **If ⊕ is the sum operation**,
> $|\Pr_u(\mathcal{G}) - \Pr_u(\mathcal{G}')| \leq \eta \left( (1 + \beta')^{T-1} \left(2 \beta'(1 - \alpha') + 4 \beta' \right) + \sum_{i=2}^{T} 2 (\beta')^i (1 + \beta')^{T-i} \right)$
>
> **If ⊕ is the mean operation**,
> $|\Pr_u(\mathcal{G}) - \Pr_u(\mathcal{G}')| \leq \eta \left( 2^{T-1} \left(4 \beta' + \frac{3(1 - \alpha')}{\alpha'} \right) + 4(T - 1)\beta' \right)$
>
> When using exact Ricci curvature, Theorem 4.4 changes only in a **constant** term, and the conclusion remains **unchanged**. Additionally, we have conducted new experiments based on this precise Ricci curvature. Our updated experiments also support our theoretical claims. Considering Ricci curvature can improve the robustness of the explanation
>
> **Table 1: Relative Error (↑) with exact Ricci Curvature (%). The ⊕ in GNN is mean operation. (Best values in bold)**
>
> | Method| Cora | Citeeseer | Pubmed | BC-OTC | BC-Alpha  | UCI  | MUTAG  | PROTEINS | IMDB-BINARY |
> |-|-|-|-|-|-|-|-|-|-|
> | GNNExplainer|1.1 | 1.1  | 2.0  | 24.9  | 12.2 | 2.1  | 2.5  | 0.3  | 0.2  |
> | GNNExplainer+Curvature   |**4.8** | **6.8** | **5.5** | **27.9** | **13.5** | **5.3** | **1.3** | **0.5**  | **0.6**  |
> | PGExplainer    | -0.6  | -0.2 |-0.7 |-0.5 |-1.7 |-1. |-0.3|-0.2| -0.9|
> | PGExplainer+Curvature              | **2.4**  | **2.4**  | **1.2** | **1.8** | **0.5** | **0.9**   | **0.9**   | **0.5**   | **0.3**  |
> | Convex                 | 23.4 | -3.6  | 4.6  | 14.0  | 2.3  | 9.5  |0.5| -4.3  | -0.5  |
> | Convex+Curvature             | **37.9** | **10.8** | **11.3** | **16.8** | **3.5** | **16.7** | **3.2** | **1.1** | **0.2** |
> | DeepLIFT               | -0.1  | 55.3 | 14.9 | -0.9   | -2.3  | 0.8  | -0.2  | 0.0  | -1.8  |
> | DeepLIFT+Curvature           | **1.7** | **67.9** | **19.8** | **0.4** | **0.5** | **1.7** | **0.3** | **2.2** | **1.0** |
> | GNNLRP                 | 20.2 | 8.9  | 10.3 | 22.9  | 22.8 | -2.2  | -0.5  | -0.7  | 0.3  |
> | GNNLRP+Curvature     | **22.4** | **13.7** | **14.2** | **23.5** | **26.6** | **3.6** | **1.7** | **0.2** | **0.8** |
> | FlowX                  | 7.0  | 6.8  | 6.8  | 0.4   | -0.6  | -1.6  | 0.6  | -0.2  | 0.1  |
> | FlowX+Curvature  | **7.6** | **7.9** | **8.1** | **1.7** | **0.2** | **0.7** | **1.1** | **0.3** | **1.3** |
>
> **Table 2: $Fidelity_{KL}$ (↓) with exact Ricci Curvature. The ⊕ in GNN is mean operation. (Best values in bold)**
>
> | Method | Cora | Citeeseer | Pubmed | BC-OTC | BC-Alpha  | UCI  | MUTAG  | PROTEINS | IMDB-BINARY |
> |-|-|-|-|-|-|-|-|-|-|
> | GNNExplainer    | 1.458| 1.361| 0.162| 0.176 | 0.099| 0.063| 0.153| 0.034| 0.241|
> | GNNExplainer+Curvature      | **1.393**| **1.372**| **0.149**| **0.175** |**0.098**| 0.063| **0.150**| **0.032**| **0.235**|
> | PGExplainer                  | **1.647**| 1.534| **0.581**| 0.209 | 0.110| 0.054| 0.156| 0.025| 0.249|
> | PGExplainer+Curvature             |1.664| **1.483**| 0.582| **0.206** | **0.109**| **0.053**| **0.146**| **0.024**| **0.243**|
> | Convex                 | 0.086| 0.005| 0.002| 0.033 | 0.007| 0.001| **0.015**| 0.001| 0.011|
> | Convex+Curvature            | **0.029**| **0.002**| 0.002| **0.030** | **0.004**| 0.001| 0.019| 0.001| 0.011|
> | DeepLIFT               | 1.012| 0.291| 0.270| 0.193 | 0.089| 0.050| 0.143| 0.040| 0.302|
> | DeepLIFT+Curvature          | **0.992**| **0.260**| 0.270| **0.192** | 0.089| **0.049**| **0.126**| 0.040| **0.301**|
> | GNNLRP                 | 0.531| 0.633| 0.193| **0.091** | **0.022**| **0.011**| **0.162**| 0.034| 0.231|
> | GNNLRP+Curvature            | **0.470**| **0.529**| **0.176**| 0.104 | 0.023| 0.013| 0.173| 0.034| **0.229**|
> | FlowX                  | 1.276| 1.164| 0.466| 0.190 | 0.081| 0.054| 0.161| 0.039| 0.253|
> | FlowX+Curvature            | **1.267**| 1.164| 0.466| 0.190 | 0.081| 0.054| **0.160**| **0.038**| **0.249**|
>
> **Table 3:Relative Error (↑) with exact Ricci Curvature (%). The ⊕ in GNN is sum operation. (Best values in bold)**
>
> | Method               | Cora | Citeeseer | Pubmed | BC-OTC | BC-Alpha  | UCI  | MUTAG  | PROTEINS | IMDB-BINARY |
> |-|-|-|-|-|-|-|-|-|-|
> | GNNExplainer         | 4.5   | -6.0  | 0.5    | 54.4   | 70.1  | 73.3  | 2.2    | 4.0  | -0.2 |
> | GNNExplainer+Curvature    | **16.0** | **2.4** | **2.6**  | **55.2**  | **70.5** | **74.7** | **5.1**   | **5.1** | **1.5** |
> | PGExplainer          | 2.7   | 6.1   | 2.6    | 3.8    | -0.3  | -0.8  | -2.8   | 0.2  | -1.2 |
> | PGExplainer+Curvature     | **4.5**  | **8.4**  | **5.6**   | **6.4**   | **1.9**  | **1.9**  | **3.9**   | **2.6** | **1.3** |
> | Convex               | 8.4   | 41.8  | 32.5   | 35.3   | 60.2  | 47.1  | 0.4    | -4.3 | 8.1  |
> | Convex+Curvature          | **15.9** | **60.6** | **37.7**  | **56.3**  | **71.2** | **58.2** | **5.6**   | **1.1** | **12.3** |
> | DeepLIFT             | 11.6  | 15.8  | 13.4   | 0.3    | -2.1  | 11.1  | -2.5   | 0.5  | -0.5 |
> | DeepLIFT+Curvature        | **16.7** | **23.9** | **14.6**  | **1.3**   | **1.9**  | **11.5** | **1.4**   | **2.1** | **1.9** |
> | GNNLRP               | 16.4  | -2.0  | 19.4   | 7.5    | 5.1   | 19.8  | -1.1   | 0.0  | -2.7 |
> | GNNLRP+Curvature          | **22.7** | **6.2**  | **23.3**  | **9.1**   | **5.7**  | **24.4** | **2.5**   | **1.5** | **2.7** |
> | FlowX    | 2.7   | 30.6  | 8.6    | 18.3   | 3.5   | 7.0   | -1.8   | -3.1 | 0.5  |
> | FlowX+Curvature | **7.8**  | **38.3** | **17.5**  | **19.3**  | **4.3**  | **11.9** | **1.3**   | **1.6** | **1.8** |
>
> **Table 4: $Fidelity_{KL}$ (↓) with exact Ricci Curvature. The ⊕ in GNN is sum operation. (Best values in bold)**
>
> | Method               | Cora | Citeeseer | Pubmed | BC-OTC | BC-Alpha  | UCI  | MUTAG  | PROTEINS | IMDB-BINARY |
> |-|-|-|-|-|-|-|-|-|-|
> | GNNExplainer | 0.961 | 0.578 | **0.549** | 0.184 | 0.245 | 0.093 | 0.177 | 0.068  | 0.284 |
> | GNNExplainer+Curvature | **0.924** | **0.518** | 0.566 | **0.175** | **0.242** | 0.093 | **0.170** | **0.067**  | **0.282** |
> | PGExplainer  | 1.762| 1.341 | **0.715** | 0.537 | 0.447 | 0.400 | 0.227 |0.048| 0.351 |
> | PGExplainer+Curvature | **1.754** | **1.339** | 0.726 |0.537| 0.447 | **0.399** | **0.225** |0.048| **0.344** |
> | Convex  | 0.353 | 0.277 | 0.075 | 0.132 | 0.009 | 0.011 | 0.010 | 0.001  | **0.017** |
> | Convex+Curvature | **0.340** | **0.273** | **0.073** | **0.053** | **0.008** | **0.010** | **0.008** |0.001| 0.021 |
> | DeepLIFT  | 1.020 | 0.481 | **0.277** | 0.178 | 0.134 | 0.202 | 0.206 | 0.054  | **0.315** |
> | DeepLIFT+Curvature  | **0.956** | **0.473** | 0.304 | **0.177** | **0.133** |0.202| **0.180** |0.054 | 0.321 |
> | GNNLRP | 0.472 | 0.589 | 0.293 | 0.174 | 0.202 | 0.234 | **0.116** | **0.033**  | **0.303** |
> | GNNLRP+Curvature   | **0.442** | **0.572** | **0.276** | **0.173** | **0.201** | 0.234 | 0.153 | 0.036  | 0.321 |
> | FlowX | 0.495 | 0.499 | 0.370 | 0.184 | 0.086 | 0.244 | **0.174** | 0.051| **0.298** |
> | FlowX+Curvature  | **0.489** | **0.470** | **0.365** | **0.182** | **0.085** |0.244 | 0.171 | 0.051| 0.301 |

---

### Official Review · Reviewer_nZXN · 2025-06-30

**Clarity:** 3
**Significance:** 3
**Originality:** 3
**Rating:** 5
**Confidence:** 4

**Summary:**

This paper defines the GNN explanation robustness problem in a fidelity view, and firstly utilizes the graph curvature property to make theorical analysis and  build a robust explanation method. Under the defined metric, the proposed method is validated to be effective under different graph tasks and on various GNN explainers.

**Questions:**

**Q1.** Following **W2**, how Euclidean distance $|x-y|$ is replaced with a CDF difference which is built only on possibility without any distance information? And how the CDFs of $\mu_{u}, \mu_{v}$, are calculated as *Figure 3*? Please make clear clarification on this part. Since later proofs are all based on this simplified calculation, due to its importance, I could only give the rejection score first. If my confusion on its correctness is properly addressed, I will raise my score to **positive at least.**

**Q2.** How the explanation subgraphs are chosen in original base explainers in *Section 6* and *D.4*? Are them selected using the same way in *Section 5*, or in a fixed amount (or fraction)?

**Ethical Concerns:**

["NO or VERY MINOR ethics concerns only"]

**Final Justification:**

Since (1) My main confusion is solved with the acknowledgement of the approximation taken in the Lemma 4.2 from the authors; (2) The experimental results on precise Ricci Curvature is additionally given, which makes the evaluation process and the underlying scientific reasoning sounder; (3) The gap between the original theoretical results and problem definition is bridged by a new analysis result, I think this paper has achieved the acceptance standard. Therefore I recommend to accept with a rating of 5.

**Limitations:**

Yes.

**Paper Formatting Concerns:**

None.

**Quality:**

3

**Strengths And Weaknesses:**

**Strengths**

**S1.** This paper firstly applies the graph curvature to analyze the GNN explanation robustness, which offers a new perspective to utilize a potential graph property to better understand GNN behavior in face of perturbation.

**S2.** With detailed theorical proof on modeling the GNN, the proposed method could be applied to different graph classification tasks and various base explainers and all achieve good performance in acceptable time cost.

**S3.** Abundant ablation studies on different experiment choices are provided in appendix, and implementation details are also provided.

**Weaknesses**

**W1.** The proof on *Lemma 4.2* is confusing (*C.1*): the first equation on $|x-y|$ introduced by **[1]** is defined on 2-D points with $\mathbb{R}\times\mathbb{R}$ coordinates so $|x-y|$ could be used to express Euclidean distance; while in this paper the graph problem is defined under a topological form, so how such mapping--"*from discrete to continuous*"-- is done should be explained. However instead, later in *C.1*. the proof directly applies $\mu_{u}$, $\mu_{v}$ from *Eq. 3*, which (1) are defined on graph node set rather than $\mathbb{R}\times\mathbb{R}$ so it’s unclear how could be used to replace on $F_X$ and $F_Y$; (2) seem not directly cumulable on a single $X$ axis as *Fig.3* since input space is not a continuous number. This leads to my further doubt on *Lemma 4.2*, which only have the node pair’s degrees as input without information from the specific structure of the graph. It seems that a denser graph (excluding edges between the node pair with their neighborhood) naturally leads to a smaller W1 distance, while this is not reflected under *Lemma 4.2*.

**[1]** Why the 1-Wasserstein distance is the area between the two marginal CDFs.

**W2.** The theoretical proof is not tightly related to the robustness definition. In the *Definition 4.1* the explanation robustness is defined as $\mathbb{E} |Pr(G)-Pr(G’)|, \mathcal{E}s \subseteq \mathcal{E}’$. Although in *Theorem 4.4* this expression is analyzed, after going through the proof in *C.2,3*, I found that the constraint $\mathcal{E}{s} \subseteq \mathcal{E}’$ is not used. In other words, this theorem holds for any perturbed graph $G’$ with no relation to explanation aspect. Later it is also found that indeed serves as a preliminary proof for *Theorem 4.5*. Instead, the *Theorem 4.5* truly contributes to the explanation robustness, motivated by line 179 “*explanation subgraph should continue to reflect the model’s behavior under perturbations to the input graph*”, with the robustness expressed as $|Pr(G_{s})-Pr(G’)|$. Though such motivation also makes sense, it’s a pity that this analysis is not truly related to the defined problem itself. I suggest in the future version, the author could consider to refine the theoretical proof to directly solve the *Definition 4.1*, maybe with some proper approximation or relaxation, or simply change the definition and metric to $|Pr(G_{s})-Pr(G’)|$, which seems also reasonable. Therefore, the scientific reasoning process of the paper could be better.

**W3.** The main performance contains few baselines, with only metric results on models with and without Curvature provided. If there’s similar defense baseline exist, it’s better to test and compare with them. If not, then I suggest the author could add some adversarial attack methods and test the performance after attack to further validate the practical effectiveness (at lease random graph noise as attack could be easily applied).

---

> ### Author Rebuttal · Authors · 2025-07-31
>
> >**Q1:The proof on Lemma 4.2 is confusing. The lemma 4.2 only has the node pair’s degrees as input without information from the specific structure of the graph.**
>
> **A1:** We thank the reviewer for carefully pointing out the concerns regarding Lemma 4.2. Lemma 4.2 was originally derived based on an approximate view, inspired by the formulation introduced in [1]. To adapt this approach for discrete graph topologies, we implicitly utilized node-degree distributions as a simplified approximation of the underlying graph structure. This approximation does not explicitly incorporate detailed graph distances. However, our results show that this method of selecting explanation subgraphs based on the approximately calculated Ricci curvature can also improve the robustness of the explanation.
>
> The Lemma 4.2 is used in lines 507-508 and 513-514 of our proofs. Motivated by the reviewer’s valuable comment, we revised these proofs by using the exact Ricci curvature, updated Theorem 4.4 accordingly. When we use exact Ricci curvature, theorem 4.4 changes only in a **constant** term, and the conclusion remains **unchanged**. The theorem 4.4 become as
>
> **If ⊕ is the sum operation**,
> $|\Pr_u(\mathcal{G}) - \Pr_u(\mathcal{G}')| \leq \eta \left( (1 + \beta')^{T-1} \left(2 \beta'(1 - \alpha') + 4 \beta' \right) + \sum_{i=2}^{T} 2 (\beta')^i (1 + \beta')^{T-i} \right)$
>
> **If ⊕ is the mean operation**,
> $|\Pr_u(\mathcal{G}) - \Pr_u(\mathcal{G}')| \leq \eta \left( 2^{T-1} \left(4 \beta' + \frac{3(1 - \alpha')}{\alpha'} \right) + 4(T - 1)\beta' \right)$
>
> Additionally, we have conducted new experiments based on this precise Ricci curvature. Our updated experiments also support our theoretical claims. Considering Ricci curvature can improve the robustness of the explanation
>
> **Table 1: Relative Error (↑) with exact Ricci Curvature (%). The ⊕ in GNN is mean operation. (Best values in bold)**
>
> | Method| Cora | Citeeseer | Pubmed | BC-OTC | BC-Alpha  | UCI  | MUTAG  | PROTEINS | IMDB-BINARY |
> |-|-|-|-|-|-|-|-|-|-|
> | GNNExplainer|1.1 |1.1 | 2.0  | 24.9  | 12.2 | 2.1  | 2.5  | 0.3  | 0.2  |
> | GNNExplainer+Curvature   |**4.8** | **6.8** | **5.5** | **27.9** | **13.5** | **5.3** | **1.3** | **0.5**  | **0.6**  |
> | PGExplainer | -0.6  | -0.2 |-0.7 |-0.5 |-1.7 |-1. |-0.3|-0.2| -0.9|
> | PGExplainer+Curvature | **2.4**| **2.4**  | **1.2** | **1.8** | **0.5** | **0.9**   | **0.9**   | **0.5**   | **0.3**  |
> | Convex | 23.4 | -3.6  | 4.6  | 14.0  | 2.3  | 9.5  |0.5| -4.3  | -0.5  |
> | Convex+Curvature  | **37.9** | **10.8** | **11.3** | **16.8** | **3.5** | **16.7** | **3.2** | **1.1** | **0.2** |
> | DeepLIFT   | -0.1  | 55.3 | 14.9 | -0.9   | -2.3  | 0.8  | -0.2  | 0.0  | -1.8  |
> | DeepLIFT+Curvature | **1.7** | **67.9** | **19.8** | **0.4** | **0.5** | **1.7** | **0.3** | **2.2** | **1.0** |
> | GNNLRP | 20.2 | 8.9  | 10.3 | 22.9  | 22.8 | -2.2  | -0.5  | -0.7  | 0.3  |
> | GNNLRP+Curvature  | **22.4** | **13.7** | **14.2** | **23.5** | **26.6** | **3.6** | **1.7** | **0.2** | **0.8** |
> | FlowX | 7.0  | 6.8  | 6.8  | 0.4   | -0.6  | -1.6  | 0.6  | -0.2  | 0.1  |
> | FlowX+Curvature | **7.6** | **7.9** | **8.1** | **1.7** | **0.2** | **0.7** | **1.1** | **0.3** | **1.3** |
>
> **Table 2: $Fidelity_{KL}$ (↓) with exact Ricci Curvature. The ⊕ in GNN is mean operation. (Best values in bold)**
>
> | Method                 | Cora | Citeeseer | Pubmed | BC-OTC | BC-Alpha  | UCI  | MUTAG  | PROTEINS | IMDB-BINARY |
> |-|-|-|-|-|-|-|-|-|-|
> | GNNExplainer    | 1.458| 1.361| 0.162| 0.176 | 0.099| 0.063| 0.153| 0.034| 0.241|
> | GNNExplainer+Curvature      | **1.393**| **1.372**| **0.149**| **0.175** |**0.098**| 0.063| **0.150**| **0.032**| **0.235**|
> | PGExplainer | **1.647**| 1.534| **0.581**| 0.209 | 0.110| 0.054| 0.156| 0.025| 0.249|
> | PGExplainer+Curvature  |1.664| **1.483**| 0.582| **0.206** | **0.109**| **0.053**| **0.146**| **0.024**| **0.243**|
> | Convex  | 0.086| 0.005| 0.002| 0.033 | 0.007| 0.001| **0.015**| 0.001| 0.011|
> | Convex+Curvature  | **0.029**| **0.002**| 0.002| **0.030** | **0.004**| 0.001| 0.019| 0.001| 0.011|
> | DeepLIFT | 1.012| 0.291| 0.270| 0.193 | 0.089| 0.050| 0.143| 0.040| 0.302|
> | DeepLIFT+Curvature | **0.992**| **0.260**| 0.270| **0.192** | 0.089| **0.049**| **0.126**| 0.040| **0.301**|
> | GNNLRP | 0.531| 0.633| 0.193| **0.091** | **0.022**| **0.011**| **0.162**| 0.034| 0.231|
> | GNNLRP+Curvature  | **0.470**| **0.529**| **0.176**| 0.104 | 0.023| 0.013| 0.173| 0.034| **0.229**|
> | FlowX  | 1.276| 1.164| 0.466| 0.190 | 0.081| 0.054| 0.161| 0.039| 0.253|
> | FlowX+Curvature  | **1.267**| 1.164| 0.466| 0.190 | 0.081| 0.054| **0.160**| **0.038**| **0.249**|
>
> **Table 3:Relative Error (↑) with exact Ricci Curvature (%). The ⊕ in GNN is sum operation. (Best values in bold)**
>
> |Method|Cora|Citeeseer|Pubmed|BC-OTC|BC-Alpha|UCI|MUTAG|PROTEINS|IMDB-BINARY|
> |---|---|---|---|---|---|---|---|---|---|
> |GNNExplainer|4.5|-6.0|0.5|54.4|70.1|73.3|2.2|4.0|-0.2|
> |GNNExplainer+Curvature|**16.0**|**2.4**|**2.6**|**55.2**|**70.5**|**74.7**|**5.1**|**5.1**|**1.5**|
> |PGExplainer|2.7|6.1|2.6|3.8|-0.3|-0.8|-2.8|0.2|-1.2|
> |PGExplainer+Curvature|**4.5**|**8.4**|**5.6**|**6.4**|**1.9**|**1.9**|**3.9**|**2.6**|**1.3**|
> |Convex|8.4|41.8|32.5|35.3|60.2|47.1|0.4|-4.3|8.1|
> |Convex+Curvature|**15.9**|**60.6**|**37.7**|**56.3**|**71.2**|**58.2**|**5.6**|**1.1**|**12.3**|
> |DeepLIFT|11.6|15.8|13.4|0.3|-2.1|11.1|-2.5|0.5|-0.5|
> |DeepLIFT+Curvature|**16.7**|**23.9**|**14.6**|**1.3**|**1.9**|**11.5**|**1.4**|**2.1**|**1.9**|
> |GNNLRP|16.4|-2.0|19.4|7.5|5.1|19.8|-1.1|0.0|-2.7|
> |GNNLRP+Curvature|**22.7**|**6.2**|**23.3**|**9.1**|**5.7**|**24.4**|**2.5**|**1.5**|**2.7**|
> |FlowX|2.7|30.6|8.6|18.3|3.5|7.0|-1.8|-3.1|0.5|
> |FlowX+Curvature|**7.8**|**38.3**|**17.5**|**19.3**|**4.3**|**11.9**|**1.3**|**1.6**|**1.8**|
>
>
> **Table 4: $Fidelity_{KL}$ (↓) with exact Ricci Curvature. The ⊕ in GNN is sum operation. (Best values in bold)**
>
> | Method               | Cora | Citeeseer | Pubmed | BC-OTC | BC-Alpha  | UCI  | MUTAG  | PROTEINS | IMDB-BINARY |
> |-|-|-|-|-|-|-|-|-|-|
> | GNNExplainer | 0.961 | 0.578 | **0.549** | 0.184 | 0.245 | 0.093 | 0.177 | 0.068  | 0.284 |
> | GNNExplainer+Curvature | **0.924** | **0.518** | 0.566 | **0.175** | **0.242** | 0.093 | **0.170** | **0.067**  | **0.282** |
> | PGExplainer  | 1.762| 1.341 | **0.715** | 0.537 | 0.447 | 0.400 | 0.227 |0.048| 0.351 |
> | PGExplainer+Curvature | **1.754** | **1.339** | 0.726 |0.537| 0.447 | **0.399** | **0.225** |0.048| **0.344**|
> | Convex  | 0.353 | 0.277 | 0.075 | 0.132 | 0.009 | 0.011 | 0.010 | 0.001  | **0.017** |
> | Convex+Curvature | **0.340** | **0.273** | **0.073** | **0.053** | **0.008** | **0.010** | **0.008** |0.001| 0.021 |
> | DeepLIFT  | 1.020 | 0.481 | **0.277** | 0.178 | 0.134 | 0.202 | 0.206 | 0.054  | **0.315** |
> | DeepLIFT+Curvature  | **0.956** | **0.473** | 0.304 | **0.177** | **0.133** |0.202| **0.180** |0.054 | 0.321 |
> | GNNLRP | 0.472 | 0.589 | 0.293 | 0.174 | 0.202 | 0.234 | **0.116** | **0.033**  | **0.303** |
> | GNNLRP+Curvature   | **0.442** | **0.572** | **0.276** | **0.173** | **0.201** | 0.234 | 0.153 | 0.036  | 0.321 |
> | FlowX | 0.495 | 0.499 | 0.370 | 0.184 | 0.086 | 0.244 | **0.174** | 0.051| **0.298** |
> | FlowX+Curvature  | **0.489** | **0.470** | **0.365** | **0.182** | **0.085** |0.244 | 0.171 | 0.051| 0.301 |
>
> [1] Why the 1-Wasserstein distance is the area between the two marginal CDFs.
>
> >**Q2:The theoretical proof is not tightly related to the robustness definition.**
>
> **A2:** Thank you for your valuable feedback. We agree that the original theoretical formulation may loosely connected to the formal definition of robustness. To address this, we have revised the theoretical analysis to ensure a tight alignment with the robustness definition. Let $r=1-\frac{|\mathcal{E}_s|}{|\mathcal{E}|}$.
>
> For theorem 4.5,
>
> **If ⊕ is the sum operation**,
> $|\Pr_u(\mathcal{G}) - \Pr_u(\mathcal{G}')| \leq \eta \left( (1 + \beta_{s})^{T-1} \left(2 \beta_{s}(1 - \alpha_{s}) + 4 \beta_{s}+2 \right) + \sum_{i=2}^{T} 2r (\beta_{s})^i (1 + \beta_{s})^{T-i} \right), s.t, \mathcal{E}_s \subseteq \mathcal{E}'.$
>
> **If ⊕ is the mean operation**,
> $|\Pr_u(\mathcal{G}) - \Pr_u(\mathcal{G}')| \leq \eta \left( 2^{T-1} \left(4 \beta_{s} + \frac{3(1 - \alpha_{s})}{\alpha_{s}} +2\right) + 2(T - 1)\beta_{s}+2r(T - 1)\beta_{s} \right), s.t, \mathcal{E}_s \subseteq \mathcal{E}'.$
>
> For theorem 4.9,
>
> $|\Pr_u(\mathcal{G}) - \Pr_u(\mathcal{G}')| \leq \eta \left( 2+2 \beta_{s}^{T} (1-r^T)+\bar{R}_s +\frac{2}{dmin(1-\gamma)}\right), s.t, \mathcal{E}_s \subseteq \mathcal{E}'.$
>
> >**Q3: The author could add some adversarial attack methods and test the performance after attack.**
>
> **A3:** Our work focuses on the robustness of graph model explanations, not on defending against graph attacks. So we do not compare with defense-related baselines, and to our knowledge, no such defense baselines exist for robust explanations.
>
> However, we do evaluate the robustness of explanations under graph perturbations (attack). For a given explanation subgraph, we randomly add and remove 10% of the edges that are not part of the explanation to create a perturbed graph G′. We repeat this process 300 times and compute the robustness score as defined in Definition 4.1. To provide a baseline, we also randomly sample some edges as explanation subgraphs, and apply the same process to get their robustness scores. We then compare the relative error between explanation method and the random baseline. The results show that our method clearly improves explanation robustness.
>
> >**Q4: How the explanation subgraphs are chosen in original base explainers?**
>
> **A4:** To ensure a fair and consistent comparison across methods, we fixed the fraction of selected edges for all explainers. Specifically, both the original base explainers and our proposed method select 10% of the total edges as the explanation subgraph. For the original explainers, the top 10% of edges are chosen based on their importance scores F(u,v) in descending order.

---

> > ### Comment · Reviewer_nZXN · 2025-08-03
> >
> > Thanks for your detailed response. With the acknowledgement of the approximation taken in the Lemma 4.2, my main confusion is solved, please remember to add this clarification in the final version. It’s impressive that besides the statement and viewpoints for taking a approximation in Lemma 4.2, a new test result with precise Ricci Curvature is also given. This is helpful to better understand its original effectiveness on robustness and makes the evaluation process sounder. It’s also delightful to see a new theoretical result is made on theorem 4.5 with tighter alignment towards the problem definition. I believe by adding the both new experimental and theoretical results, this paper would achieve a strong quality in the final version. Therefore, I’ll raise my score to 5 now.

---

> > > ### Author Response · Authors · 2025-08-04
> > >
> > > Thank you for taking the time to review our rebuttal. We will include the clarification of Lemma 4.2 and revise Theorems 4.5 and 4.9 in the final version. We deeply thank your reading, understanding, and appreciating of our rebuttal!

---

### Official Review · Reviewer_VzwU · 2025-07-01

**Clarity:** 4
**Significance:** 3
**Originality:** 4
**Rating:** 4
**Confidence:** 3

**Summary:**

The paper “Robust Explanations of Graph Neural Networks via Graph Curvatures” tackles the problems of non-faithfulness explanation of retraining method and the contradiction and uncertainty explanations of ensemble method, by introducing Ricci curvature and effective resistance in graph theory as robust score to obtain more robust explanation. It presents theoretical results on the relation of the explanation robustness to Ricci curvature and effective resistance, and an objective combining Ricci curvature and effective as robust score is proposed to achieving the trade-off between fidelity and robustness. The experimental results on nine datasets demonstrates the effectiveness of the proposed method.

**Questions:**

Please refer to weaknesses

**Ethical Concerns:**

["NO or VERY MINOR ethics concerns only"]

**Final Justification:**

Since the authors address the clarity concerns by the answers of questions 1 and 4, I raise the clarity score to 4. However, I still think the relations and differences between Ricci curvature and effective resistance should be more discussed and investigated.

**Limitations:**

yes

**Quality:**

3

**Strengths And Weaknesses:**

**Strengths**
1. The motivation of using Ricci curvature and effective resistance as robust score is clear, and the theorems is interesting and novel

2. The theoretical analysis of the relation of robust explanation to Ricci curvature and effective resistance gives valuable insight
The ablation experiment results are convincing

**Weaknesses**
1. Some presentations remain unclear, e.g., what is the size of the explanation sub-edges E_s, how the determine this size

2. Is there any connection between Ricci curvature and effective resistance, can we combine both of the score in the final objective to further boost performance, it is better to provide such discussion

2. The baseline only includes the basic GNN-explaining methods, it is better to evaluate the performance on methods like ensemble method and retraining method

3. What does the “+Curvature” mean in the effective resistance based results on Table 3 and 4, is that a typo? And the base result (e.g. GNNExplainer) in Table 1 cannot align with Table 3, what are the differences here

---

> ### Author Rebuttal · Authors · 2025-07-30
>
> Thank you for taking the time to review our paper!
> >**Q1: what is the size of the explanation sub-edges E_s**
>
> **A1:** Thank you for the question. In our experiments, the size of the explanation subgraph E_s is set to 10% of the total number of edges in the input graph. This fixed ratio allows for fair comparison across different methods and ensures the explanation remains concise.
>
> >**Q2: Is there any connection between Ricci curvature and effective resistance, can we combine both of the score in the final objective to further boost performance?**
>
> **A2:** Thank you for the insightful suggestion. We conduct a statistical analysis on three datasets using both Spearman and Pearson correlation coefficients, and find no significant correlation between Ricci curvature and effective resistance. The results are shown  in the table.
>
> Table 1: The correlation coefficient and p-value
> | Datasets            | Pearson | p-value | Spearman | p-value |
> |---------------------|---------|---------|----------|---------|
> | PubMed              | -0.033  | 0       | -0.023   | 0       |
> | Coauthor-Computer   | 0.055   | 0       | 0.067    | 0       |
> | Coauthor-Physics    | 0.022   | 0       | 0.028    | 0       |
>
> In the theory section, we analyze how Ricci curvature and effective resistance are individually related to model robustness and explanation robustness. However, we do not analyze their joint effect on robustness. Therefore, in our final objective, we treat these two metrics separately.
>
> We think that combining both scores into a unified objective does not always lead to better performance. For example, in some cases, selecting edges A, B, and C based on effective resistance provides more robust explanations, while in the same cases, Ricci curvature favors edges D, E, and F. These conflicting preferences may reduce overall robustness. We acknowledge this as an interesting direction, and plan to explore it further in future work.
>
> >**Q3:The baseline only includes the basic GNN-explaining methods, it is better to evaluate the performance on methods like ensemble method and retraining method.**
>
> **A3:** Thank you for the helpful suggestion. Regarding the robustness of graph model explanations, few works use ensemble or retraining methods to improve explanation robustness. In other fields, such as computer vision, such methods have been used to improve robustness, but they are not commonly applied in the graph domain.
>
> In this work, our proposed method is both model-agnostic and explanation-method-agnostic. It identifies robust edges based on the geometric properties of the graph data. For this reason, we focus our evaluation on standard GNN explanation methods and show that our approach can enhance basic GNN-explaining methods robustness. We appreciate the suggestion and consider evaluating the performance on methods like ensemble method and retraining method for future work.
>
> >**Q4: What does the “+Curvature” mean in the effective resistance based results on Table 3 and 4, is that a typo? And the base result (e.g. GNNExplainer) in Table 1 cannot align with Table 3, what are the differences here?**
>
> **A4:** Thank you for pointing this out. The “+Curvature” in Tables 3 and 4 is indeed a typo. We apologize for the confusion and the poor reading experience it may have caused. Regarding the base results, although the same dataset and method are used in Table 1 and Table 3, the target nodes/edges/subgraphs to be explained are different. As a result, the reported results are not directly aligned.

---

> > ### Comment · Reviewer_VzwU · 2025-08-09
> >
> > Thank you for your replies, my concerns are addressed, and I have raised the clarity score to 4.

---

### Note · Authors · 2025-08-14

We sincerely thank all reviewers and the AC for their time, effort, and constructive feedback during the review and discussion phases.

We are pleased to note that Reviewer VzwU’s concerns have been addressed. For reviewer nZXN, we have clarified the approximation taken in Lemma 4.2, and, in addition, we have incorporated both a new experimental result using the exact Ricci curvature and an enhanced theoretical result for Theorem 4.5 with tighter alignment to the problem definition. These changes have not only resolved the main confusion but also strengthened the soundness of the evaluation process and the theoretical contribution. For reviewer Eh2L, we have conducted statistical analysis to evaluate the robustness of our results. Besides, we also test the robustness of both our method (based on Ricci curvature) and the original explanation method on several datasets, using different random seeds.

Overall, the review process has substantially improved the manuscript’s clarity,  and theoretical soundness. We are grateful for the reviewers’ time and effort, which have significantly improved the quality of the paper.

---

### Decision · Program_Chairs · 2025-09-17

**Decision:**

Accept (poster)

**Comment:**

The paper introduces a novel approach to enhance GNN explanation robustness using graph geometric properties, specifically Ricci curvature and effective resistance.  All reviewers acknowledged the novelty of applying Ricci curvature and effective resistance to GNN explanation robustness, with theoretical proofs establishing upper bounds on robustness. The method was validated across 9 datasets spanning node classification, link prediction, and graph classification tasks.  The approach can be applied to various base explanation methods, demonstrating broad applicability.

Initially, two reviewers have concerns on Approximation in Lemma 4.2, Statistical significance , Global vs. local lambda parameter  and Tight connection to robustness definition. All reviewers who engaged post-rebuttal were satisfied with the responses, with nZXN explicitly raising their score and VzwU addressing all concerns.

While this is a solid technical contribution with novel theoretical insights, it represents an incremental advance in GNN explainability rather than a breakthrough result.  I tis good to present in poster.